# Numerical modelling of the evolution of a river reach with a complex morphology to help define future sustainable restoration decisions

Rabab Yassine[1,2,3*], Ludovic Cassan[1], Hélène Roux[1], Olivier Frysou[3], and François Pérès[2]

[1]Institut de Mécanique des Fluides de Toulouse (IMFT), Université de Toulouse, CNRS, Toulouse, France
[2]INP/ENIT, LGP, Université de Toulouse, Tarbes, France
[3]Pays de Lourdes et des Vallées des Gaves, Lourdes, France
[*]Currently at EGIS Business Unit: Major Structures, Water, Environment, Energy, Montpellier, France
**Correspondence:** Rabab Yassine (rabab.yassine@egis-group.com)

**Abstract.** The evolution of river morphology is very complicated to predict, especially in the case of mountain and Piedmont rivers with complex morphologies, steep slopes, and heterogeneous grain sizes. The "Lac des Gaves" (LDG) reach, located within the Gave de Pau river in the Hautes-Pyrénées department, France, has precisely the complex morphological characteristics mentioned above. This reach has gone through severe sediment extractions for over 50 years, leading to the construction of
two weirs for riverbed stabilisation. Two large floods resulted in changes in the LDG's hydromorphological characteristics as it went from a single channel river section to a braided river reach. In this study, a 2D hydromorphological model is developed with the TELEMAC-MASCARET system to reproduce the evolution of the channel following a flood that occurred in 2018. The model's validity is assessed by comparing the simulated topographic evolution to the observed one. The results reveal the challenge to choose well-fitted sediment transport equations and friction laws that would make it possible to reproduce such complex morphology. Even if the exact localisation of the multiple channels forming the braided nature of the LDG
was challenging to reproduce, our model could provide reliable volumetric predictions as it reproduces the filling of the LDG correctly. The influence of the two weirs on the river's current and future morphology is also studied. The aim is to provide decision-makers with more reliable predictions to design suitable restoration measures for the LDG reach.

## 1   Introduction

Flood events can lead to considerable sediment transport that has an influence on flow dynamics. Understanding the interactions between flow dynamics and morphological changes is thus of growing interest in the research community (Guan et al., 2015), especially in mountainous regions where the interactions between water and sediments are complex. Rickenmann et al. (2016) highlighted the critical influence of sediment transport during flood events in alpine catchments and the inherent damages. Reisenbüchler et al. (2019a) showed that morphodynamics could increase the flood intensity leading to more dramatic conse-
quences. This is particularly true in mountainous catchments where the important sediment supply from the upstream torrents and torrential rivers may expose the downstream fluvial system to great danger during flood episodes and increase the related damages (Reid et al., 2007; Badoux et al., 2014). For instance, channel conveyance capacity can decrease when consequent amounts of sediments are deposited within the riverbed, increasing river diversion risks toward surrounding areas (Badoux

et al., 2014; Recking et al., 2012; Reid et al., 2007; Rickenmann et al., 2016; Rinaldi and Darby, 2007). Understanding sediment transport and especially bedload is thus essential for establishing a coherent flood control plan and defining sustainable restoration strategies (Kang and Yeo, 2015). Besides safety issues, bedload transport, combined with water discharge, is considered a fundamental driver of river morphodynamics and risks of overflowing. They can affect habitat, aquatic ecosystems, river stability, and natural hazards (Wohl et al., 2015).

River restoration for flood prevention purposes is generally related to achieving a sufficient degree of protection through the design of solutions ranging from the installation of physical infrastructures to alternative measures for risk reduction (Arnaud-Fassetta et al., 2009). Reliable numerical modelling of flow and sediment dynamics with a good field expertise can be useful in this case for better river management. Numerical models can provide quantified answers on the configuration of flows during a flood event, which can be challenging to measure on the ground (Chapuis, 2012). Morphological models coupled to hydrodynamic ones (Reisenbüchler et al., 2019a) have now been applied to various rivers of different sizes and characteristics to examine the evolution of alluvial river channels (Carr et al., 2015; Guan et al., 2015, 2016; Ham and Church, 2012; Rinaldi and Darby, 2007; Tal and Paola, 2010; Tu et al., 2017; Ramirez et al., 2020). This means that the hydrodynamic model provides information on the turbulence, shear stress, and flow to the morphological model that uses it to compute sediment transport rates and bed evolution (*i.e.* erosion and deposition rates). Simultaneously, the morphological modifications have then an influence on the hydrodynamic simulation. However, sediment transport rates are usually calculated with empirical formulas (Meyer-Peter and Müller, 1948; van Rijn, 1984a; Einstein, 1950; Wilcock and Crowe, 2003) mostly derived from laboratory experiments with numerous simplifications of real field conditions although more recent formulas are partially based on field data (Recking, 2013a; Lefort, 2007). To have a physically realistic simulation, it is necessary to provide the model with realistic bedload transport rates to introduce reliable boundary conditions and physical modelling within the study area. Besides, the morphological processes occurring in the field are often simplified. This is why a field investigation and scientific monitoring before developing the model must be very well conducted to help the model operator criticise and improve its predictive abilities.

When the model is well-calibrated and validated enough on real field data, the main advantage of modelling is that it is possible to simulate restoration scenarios, challenging to implement in the field (Arnaud, 2012). Two-dimensional (2D) numerical models are increasingly being used for flood modelling and river management in general. The majority of these models consider the resolution of Shallow Water Equations (SWE) (Hervouet, 2003). As the impacts of morphological modifications on flow dynamics can be considerable, considering sediment transport is of primary importance when the purpose is to design sustainable restoration solutions. Morphodynamic simulations are thus required to represent bed evolution following the implementation of a restoration measure, especially in Piedmont rivers, where these factors can highly influence hydrodynamics. Numerical models allow considering complex geometries with several channels and various classes of sediments. For instance, they can provide information about the velocity and the suspended concentration of transported sediments, which has to be known for ecological purposes. They can estimate the time scale of erosion or deposition for flood impact forecasts. They can also evaluate morphological evolution in areas lacking expertise, for instance, close to hydraulic structures with a specific design.

The effects of the interactions between hydrodynamics and morphodynamics have proved to be particularly dramatic during the flood of 2013, an almost 100-year return period event, that severely impacted the "Gave de Pau" catchment, especially the "Lac des Gaves" reach in the Hautes-Pyrénées department in France that we will name LDG in this article. This former artificial lake within the "Gave de Pau"'s riverbed, delimited by two weirs, has undergone years of sediment extractions. These activities lead to a robust hydromorphological imbalance that is disturbing the watercourse's normal functioning in this area. Today, after the flood of June 2013, the lake is almost completely filled with sediments, which may lead to river diversion towards populated areas. Upstream the second weir, the "Gave de Pau", has precisely the complex morphological properties mentioned above. In this area, the river presents specific aspects of Piedmont rivers, characterized by very heterogeneous grain sizes and a complex braided morphology, which indicates considerable sediment delivery from the upstream catchments. On the opposite, downstream the weir, an active channel shrinkage is observed, characteristic of a sediment deficit and a sediment discontinuity that led to serious ecological damages and navigation problems.

The TELEMAC-MASCARET[1] modelling system has been considered well suited to perform 2D morphodynamic simulations on the LDG reach. Indeed, previous studies have shown that TELEMAC/Sisyphe was able to reproduce processes of erosion/deposition accurately in similar configurations (Reisenbüchler et al., 2020, 2019b; Cordier et al., 2019). Sisyphe enables the use of different transport formulas (Meyer-Peter and Müller, 1948; van Rijn, 1984b) and also take into account various factors influencing sediment transport, such as the effect of the bed slope (Koch and Flokstra, 1981; Soulsby, 1997) on the magnitude of the bedload transport (Riesterer et al., 2016). It also offers the possibility of programming other formulas, both for the parameterisation of friction and for solid transport, a possibility which has been used here to introduce formulations more adapted to the context of mountain rivers. However, it is necessary to note that this type of calculation has been little explored on such complex morphologies specific to Piedmont rivers. Most simulations considering sediment transport with this model have been carried out on laboratory cases or real case studies with lower slopes and/or simpler morphologies (Lepesqueur et al., 2019; Orseau et al., 2021). For braided morphodynamic modelling, the model performance can be provided by a specific indicator a the scale of the area of interest (Williams et al., 2013, 2016a, b; Rifai et al., 2014; Gonzales de Linares et al., 2021). Thus, it is interesting to evaluate the model's performance on such kind of complex morphology.

The present work serves to illustrate: (1) the ability of a 2D numerical model to reproduce hydromorphological processes in complex river morphology, (2) the performance of different friction laws and sediment transport equations, and (3) how a 2D hydromorphological model can help river managers to better understand the dynamics within the LDG reach in order to evaluate the impacts of a given restoration measure on the system and to adopt a sustainable and rational management orientation (De Linares, 2007).

The paper is organized as follows: section 2 introduces the study site and its characteristics. Section 3 describes the model with an emphasis on the friction laws and bedload formula. Section 4 presents the methodology to implement the hydromorphological model on the LDG area as well as the performance evaluation. The results are detailed and analyzed in section 5, with special attention to the sensitivity of the simulated behaviour with respect to the friction laws and the bedload formula. The main findings are summarized in section 7.

---

[1]http://www.opentelemac.org/

## 2 Study area

### 2.1 The "Gave de Pau" catchment

The "Gave de Pau" watershed (Fig. 1) is located in the western Pyrenees between the lowland of Lourdes (420 m asl) and the Spanish border in the south, where the highest French Pyrenean peaks culminate (Vignemale 3298 m asl, Taillon 3144 m asl). The "Gave de Pau" River originates in the well-known "Cirque de Gavarnie" around 2600 m asl (UNESCO World Heritage). The upstream part of the catchment has typical mountainous characteristics described by steep slopes, important sediment transport, high-water seasons observed between the end of spring and the beginning of summer, and a very dense hydrographic

network. The two rivers constituting the "Gave de Pau" main stream are the "Gave de Gavarnie" (right bank) and the "Gave de Cauterets" (left bank). In high flow seasons, these two watercourses showed that they could transport significant amounts of sediments. They are thus considered as the primary sources of sediments coming from the upstream part of the catchment and deposited in the downstream central valley of Argelès-Gazost, where the LDG and most of the stakes are located (Fig. 2).

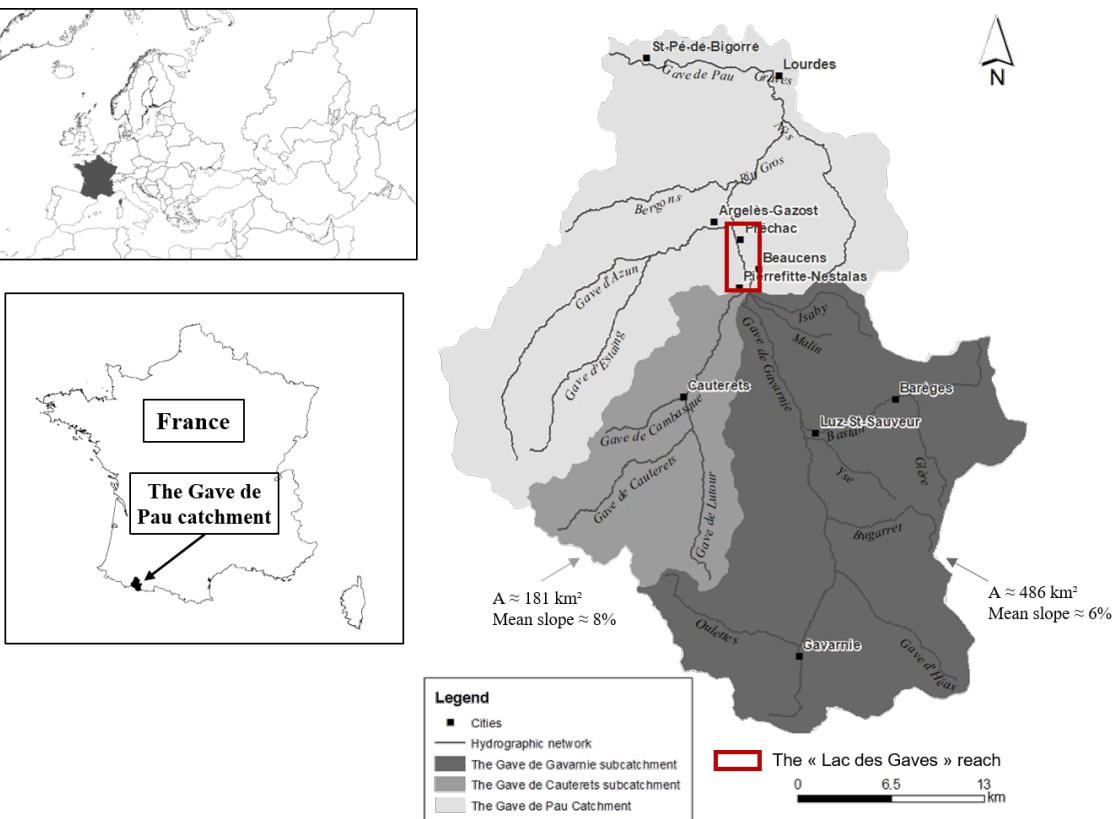

**Figure 1.** The "Gave de Pau" catchment and its main upstream sub-catchments: the "Gave de Cauterets" and the "Gave de Gavarnie" sub-catchments (A is the drainage area and the mean slope of the watercourses is also introduced). The LDG reach is represented by the red rectangle between the cities of Pierrefitte-Nestalas and Argelès-Gazost, Hautes-Pyrénées, France

### 2.1.1 The Gave de Cauterets subcatchment

The Gave de Cauterets subcatchment has a compact shape and besides the Cambasque (left bank) and Lutour (right bank) rivers its other tributaries are very small and they have a typical torrent morphology with very steep slopes. The main valley and its tributary valleys have a South-North orientation. In these conditions, we may expect a simultaneous hydrological functioning of its main tributaries in case of widespread precipitations and thus a rapid concentration of overland flow as soon as we reach the city of Cauterets. From the Spanish border (south), where its main tributaries originate, to its outlet, the drainage slope is rarely below 2% and it reaches its maximum in the gorges areas. The mean elevation of the watershed is above 2000 m asl and two thirds of its surface is between 1500 and 2500 m asl.

### 2.1.2 The Gave de Gavarnie subcatchment

Before its junction with the Gave de Cauterets, the Gave de Gavarnie drains a very wide catchment whose area is approximately 486 $km^2$. From its origin (Cirque of Gavarnie) to its outlet, its main stream is supplied by active torrents (the Gave d'Héas: 75 $km^2$, the Yse torrent: 13.5 $km^2$ and the Bastan torrent: 100$km^2$). The Bastan torrent was the tributary that showed the most impressive sediment transport activity during the flood of 2013. Today, it is considered as the main contributor in terms of sediment supply to the Gave de Gavarnie and even the downstream valley.

As for the Gave de Cauterets and its main tributaries, the Gave de Gavarnie valley has a South-North orientation. Unlike the Gave de Cauterets catchment with its compact morphology, the Gave de Gavarnie watershed is wider and the supplies of its main tributaries are gradually distributed from upstream to downstream. It is thus very probable that these physiographic factors condition an uneven distribution of the precipitations at the catchment scale. During rainy events, we may expect contrasted repartitions from a tributary to another. Therefore, different hydrological situations may sometimes lead to the same discharge and volume at the outlet.

The main morphometric characteristics of each subcatchment are presented in table 1. Catchment characteristics were obtained thanks to a GIS analysis of several spatial local databases (BD ORTHO®, BD ALTI®, Corine Land Cover®, IGN©).

## 2.2 The LDG reach

The LDG (Fig. 2) is an artificial lake located in the main stream of the "Gave de Pau" river. Like many rivers and lakes worldwide, it has gone through very intensive sediment extractions estimated to be around four million cubic meters over the past century. These activities led to constructing two weirs, one upstream and one downstream of the lake, to stabilize the riverbed. The large flood of 2013 highlighted a critical amount of impairments at the catchment scale, especially within this reach. It showed that the lake is now acting like a sediment trap blocking all the sediments coming from the upstream mountain streams. A brutal longitudinal profile discontinuity is observed and leads to an increased risk of river diversion towards populated areas, destruction of hydraulic structures' foundations, shrinkage of the active channel, a global incision

**Table 1.** Main characteristics of the Gave de Cauterets and the Gave de Gavarnie subcatchments

|  | The Gave de Cauterets | The Gave de Gavarnie |
|---|---|---|
| Drained area (km²) | 181 | 486 |
| Perimeter (km) | 79 | 121.1 |
| Max. elevation (m asl) | 3260 | 3260 |
| Mean elevation (m asl) | 1961 | 1905 |
| River source elevation (m asl) | 2500 | 2600 |
| Outlet elevation (m asl) | 457 | 457 |
| Elevation of the highest peak (m asl) | 3300 | 3300 |
| Main channel length (km) | 28.1 | 38.6 |
| Mean slope (%) | 8 | 6.4 |
| Mean annual discharge (m³/s) | 5.5 | 21 |

(more than 3 meters), to name a few. This is mainly due to the LDG reach position, which is immediately located after the junction of the two mountain streams presented above.

## 2.3 Flood events

Like several research (Blanpied, 2019) and engineering projects (IDEALP, 2014; PLVG, 2015; SUEZConsulting, 2019), this work was initiated by the exceptional flood of June 2013 that had a very strong impact on the entire central Pyrenees. This extreme event was caused by heavy rainfall combined with rapid and abundant snowmelt due to a brutal increase in temperature after a very cold spring. The peak discharge was estimated to be about $742\ m^3/s$ in Lourdes, corresponding to a 100-year return period flood compared to compared to the monthly averaged discharge of $90\ m^3/s$ (DREAL-Midi-Pyrénées, 2013; PLVG, 2015). Besides the two casualties and the catastrophic material damage estimated at nearly 300 million of euros, this event has demonstrated the major influence of sediment transport in the hydromorphological dynamic of the catchment's streams. In fact, the extreme hydrology combined to a very high rate of sediment delivery from the upstream catchments exposed the downstream fluvial system to great danger in terms of very important sediment depositions, serious bank erosions that caused the collapse of roads and buildings, destruction of hydraulic structures' foundations and significant ecological damages (Fig. 3).

During this event, the LDG acted like a sediment trap as it intercepted almost all the sediments coming from the upstream catchment (Fig. 2 and 4). Its morphology completely changed as it went from a lake/single-channel river section to a braided river reach. Five years after the flood of June 2013, another highly morphogenetic but of lesser magnitude flood occurred in June 2018. The peak discharge was estimated to be about $332\ m^3/s$ corresponding to a 10-year return period. Even if the damages are not comparable to the ones caused by the flood of 2013, the 2018 flood event greatly impacted the morphology of all the watercourses of the Gave de Pau catchment and exacerbated the filling phenomenon in the LDG. Today, the lake is almost completely filled, and avulsion risks are observed as the left bank elevation is lower than the bed elevation.

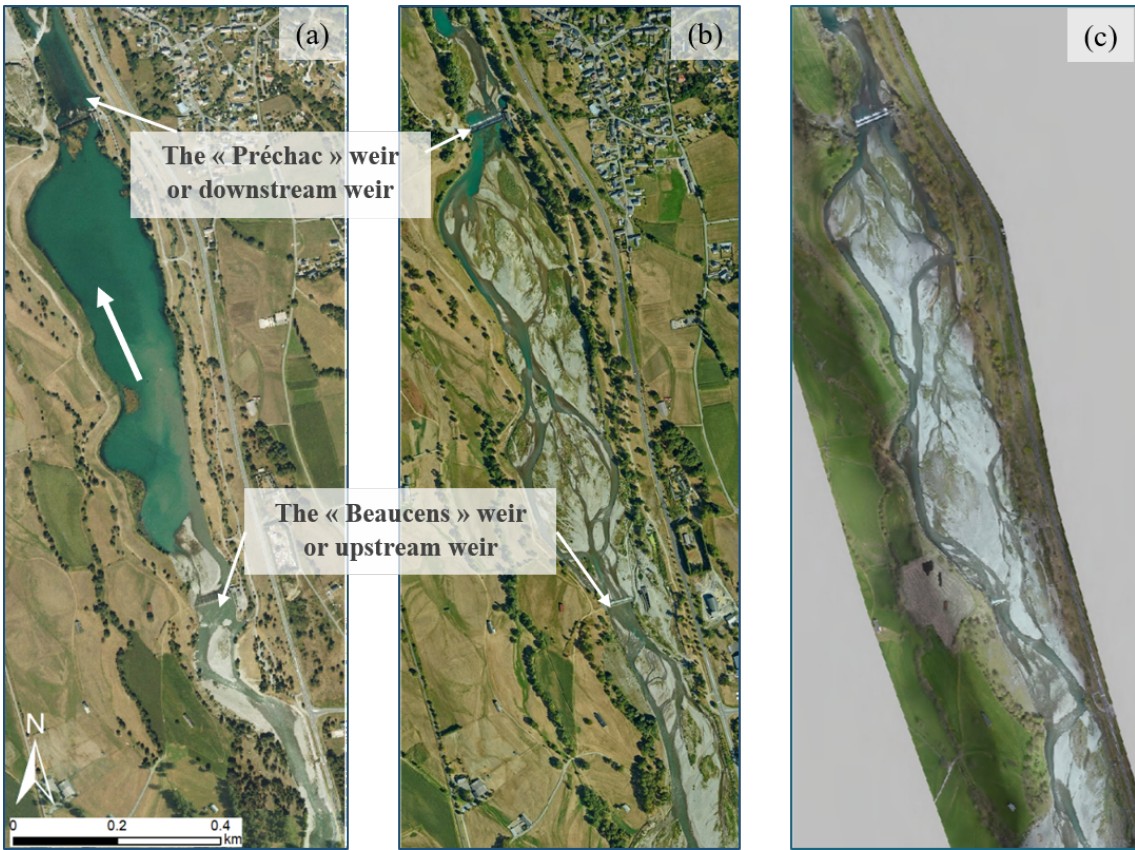

**Figure 2.** Morphological changes observed in the LDG reach. Aerial photo in (*a*): 2006, (*b*): 2016 after the flood of June 2013 and (*c*): 2019 after the flood of June 2018 (source: IGN BD ORTHO, PLVG)

## 2.4 Restoration implications

To reestablish the natural flow, reduce flood risks, and restore the ecological continuity, river managers are considering lowering or even suppressing the weirs. However, even if these restoration measures seem to be relevant over the long term, many hydromorphological and ecological effects might emerge, such as backward erosion, over-delivery of sediments to the downstream fluvial system, to name a few (Malavoi et al., 2011). Besides, due to mining activities during almost three decades (1941-1969) immediately upstream the LDG reach, there are still many excavations and waste on the former plot that could,

in some instances, be dangerous from a human or environmental point of view. Today, these metallic residues (zinc, mercury, lead, arsenic) are suspected of accumulating in living beings (fish, mosses, invertebrates) or fixed in the fine fraction of the alluvial stock (sediments of the type: clay, silt). Most of these sediments are now suspected to be stored in the LDG, located approximately 5 km downstream from the former mining site. Thus, depending on which restoration measure is selected, this information must be considered knowing the risks of contamination of the downstream area.

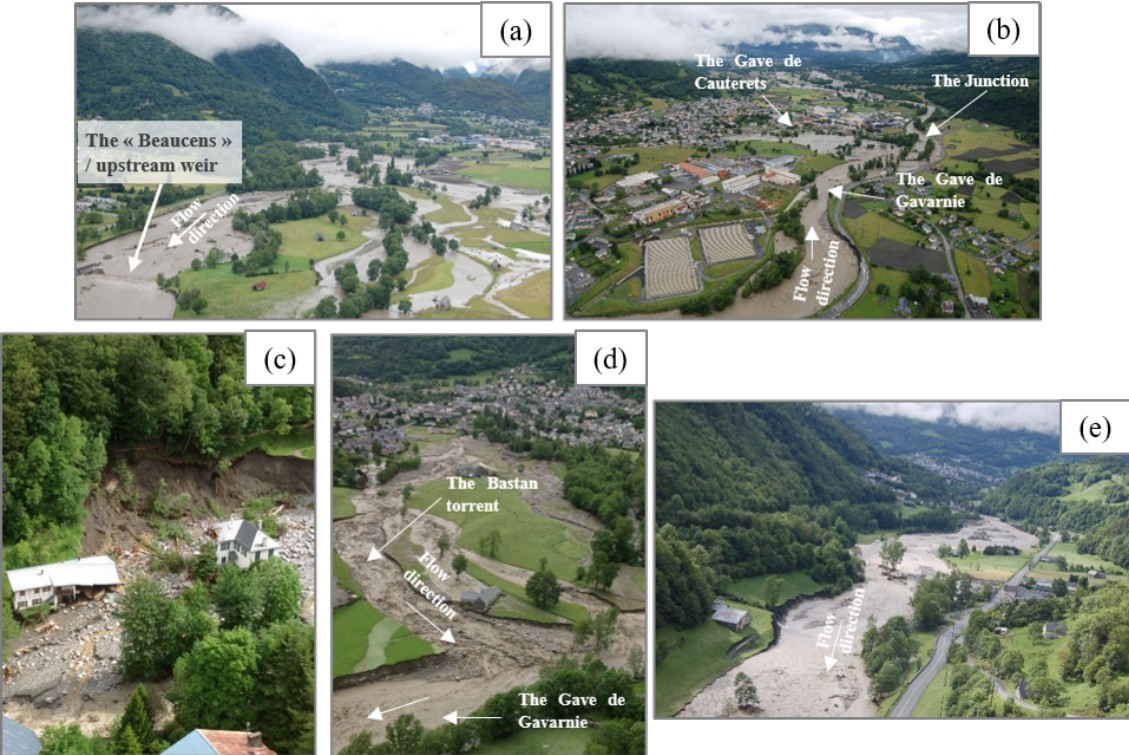

**Figure 3.** Some examples of damages caused by the flood of June 2013 at different locations and different streams. (*a*) The upstream part of LDG reach during the flood. This photo illustrates the river avulsion that occurred; the riverbed created few channels in the Adast plain: left bank (Upstream (south) view) (DDT65and PLVG), (*b*) the "Gave de Gavarnie" and "Gave de Cauterets" at their junction. During the event, the erosion of the "Gave de Gavarnie's" right bank destroyed the main access road, and the villages of Pierrefitte and Soulom were flooded by the "Gave de Cauterets" (DDT65 and PLVG), (*c*) significant erosions and destruction of buildings located in the "Gave de Cauterets" active channel (RTM65), (*d*) the "Bastan" torrent and the "Gave de Gavarnie" at their junction. This photo illustrates the important morphological activities that were engaged during the flood at this location (RTM65), (*e*) the "Gave de Gavarnie" at the Saligos plain immediately after its junction with the "Bastan" torrent. Significant bank erosions are observed in this area (DDT65 and PLVG)

A hydro-morphological 2D model was developed at the LDG reach scale to understand the different morphological processes within this channel and help river managers make an informed decision on the restoration of this reach. One of the processes on which the modeling efforts will focus is the deposition phenomenon within the LDG as it represents the potential volumes that might be mobilised if the weir lowering/removal restoration measure is considered.

## 3    Model description

The system TELEMAC-MASCARET is considered for the numerical simulations. TELEMAC-MASCARET is an open-source software package with numerous modules to compute free surface flows, sediment transport, swell, and water qual-

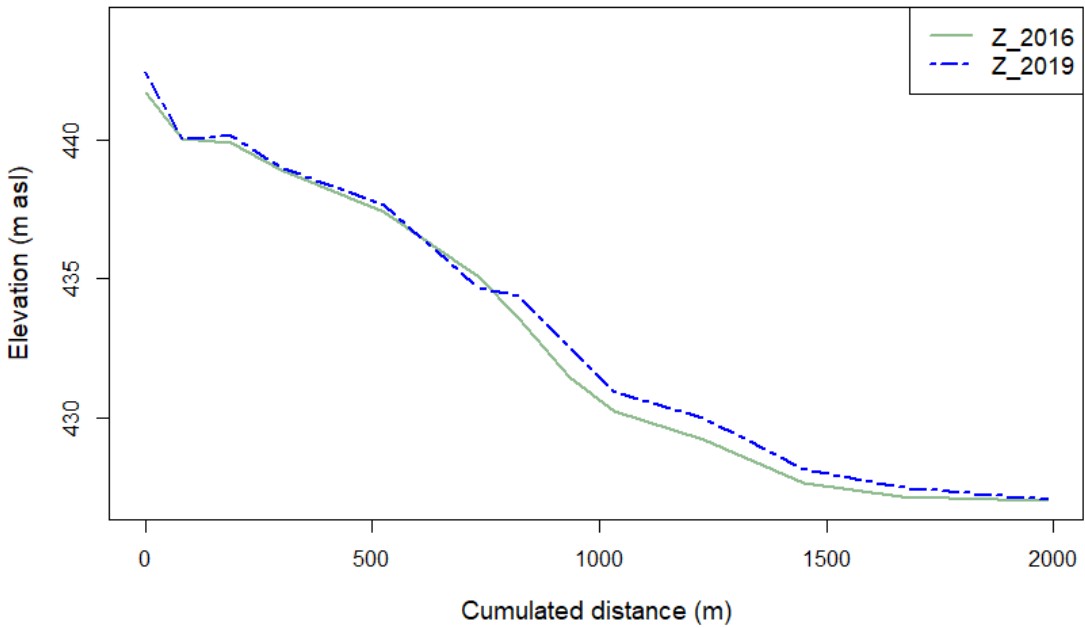

**Figure 4.** Longitudinal evolution of the Lac des Gaves reach following the flood of 2018.

ity (Hervouet, 2003). Among these modules, we selected the ones related to hydrodynamic and morphodynamic processes: TELEMAC2D and SISYPHE.

In this section, the hydrodynamic module is introduced as well as its morphodynamic module SISYPHE. Then its application
to the study area is presented. Finally, the model's performance will be assessed, and we will conclude on the difficulties encountered while performing the simulation on such complex morphology.

### 3.1 Hydrodynamic module

The hydrodynamic module, TELEMAC2D, solves Shallow Water Equations (SWE) simultaneously (de Saint-Venant, 1871) (Eq. 1).

$$
\begin{cases}
\partial_t h + \boldsymbol{u}.\nabla(h) + h\nabla.(\boldsymbol{u}) &= 0 \\
\partial_t u + \boldsymbol{u}.\nabla(u) &= -gd_x z_f - gS_{f,x} + h^{-1}\nabla.(hv_t\nabla u) \\
\partial_t v + \boldsymbol{u}.\nabla(v) &= -gd_y z_f - fS_{f,y} + h^{-1}\nabla.(hv_t\nabla v)
\end{cases}
\tag{1}
$$

where t [s] is the time, $\partial_t = \partial/\partial_t$, $\nabla = (\partial_x, \partial_y)$ is the gradient vector field, $g = 9.81\ m/s^2$ the gravitational acceleration, $h$ [m] is the water depth, $\boldsymbol{u} = (u, v)$ [m/s] is the depth-averaged flow velocity vector with $u$ and $v$ [m/s] the components along the

longitudinal x-axis and transversal y-axix direction respectively, with $|\boldsymbol{u}|$ [m/s] the module of $\boldsymbol{u}$, and $v_t$ $[m^2/s]$ is the turbulent eddy viscosity term.

The TELEMAC model treats turbulence from a diffusion term. Four options are available and they were all tested in the framework of the study:

- the constant viscosity model. The associated coefficient represents molecular viscosity, turbulent viscosity and dispersion;

- the Elder model. This model takes into account the dispersion by assuming that the vertical profiles of the velocities are logarithmic;

- the k-Epsilon model. This model solves the transport equations for k (the turbulent energy) and Epsilon (turbulent dissipation). The latter is known to be more expensive in terms of computational time and requires a finer mesh compared to the other models;

- the Smagorinski model, which is generally used for maritime domains with large-scale fluctuation phenomena.

## 3.2 Sediment transport and bed evolution module

The morphodynamic module is based on the Exner equation (Eq. 2) (Exner, 1920), which can be coupled with the equation of the hydrodynamic module:

$$(1-n)\frac{\partial Z_f}{\partial t} + \nabla.Q_s = 0 \tag{2}$$

where $n$ is the non cohesive bed porosity [-], $Z_f$ [m] corresponds to the river bottom elevation, and $Q_s$ $[m^2/s]$ the bedload rate per unit width. Further information on this module can be found in Tassi and Villaret (2014)).

### 3.2.1 Friction laws

Two friction laws were considered: the widely known Manning-Strickler (1923) formula (Eq. 4) and the Ferguson (2007) formula. The Ferguson (2007) friction law has been proposed to ensure the transition between a uniform profile related to relative shallow depths and larger relative depths, whereas the Manning-Strickler formulation is better suited for larger relative depths. This equation of Ferguson (2007) has been tested on a wide range of data and has proved to be efficient to cover all hydraulic configurations encountered from headwaters to lowland rivers. It is expressed as follows (Eq. 3):

$$\frac{U}{\sqrt{gR_hS}} = \frac{2.5\frac{R_h}{D_{84}}}{\sqrt{1+0.15\left(\frac{R_h}{D_{84}}\right)^{5/3}}} \tag{3}$$

with $S$ [m/m] the river bed slope, $D_{84}$ [m] diameter for which 84% of sediments are finer, $R_h$ (m) the hydraulic radius, $U$ $[ms^{-1}]$ the mean flow velocity, and $g$ $[ms^{-2}]$ the gravitational acceleration.

The Ferguson law uses the $D_{84}$ as a proxy of the bed roughness (Ferguson, 2007). value of $D_{84}$ is directly obtained thanks to the grain-size measurements done at the bed surface in the LDG area.

The Manning-Strickler friction law can be expressed as follows (Eq. 4):

$$U = K R_h^{3/2} S^{1/2} \qquad (4)$$

with $U$ $[ms^{-1}]$ the mean flow velocity, $S$ [m/m] the river bed slope, $R_h$ [m] the hydraulic radius, and $K$ $[m^{1/3}s^{-1}]$ the friction

coefficient.

### 3.2.2 Bedload transport formulas

The morphodynamic module SISYPHE considers several semi-empirical sediment transport formulas (Tassi and Villaret, 2014). The module also offers the possibility to code a formula, if it is not included. In our case, we considered two bed-load transport formulas: the Meyer-Peter and Müller (1948) formula (Eq. 5) and the Recking (2013b) formula (Eq. 6).

The Meyer-Peter-Müller formula:

The Meyer-Peter-Müller equation is a threshold equation and its original formulation considers a critical Shields parameter equal to 0.047. A sensitivity analysis was performed on this parameter as its value can highly influence sediment transport. The formula is written as follows (Eq. 5):

$$\Phi = 8 \left[ \left( \frac{K'}{K} \right)^{3/2} \tau^* - 0.047 \right]^{3/2} \qquad (5)$$

$\Phi$ is the dimensionless solid transport, calculated as $\Phi = \frac{q_{sv}}{\sqrt{g(\rho_s/\rho - 1)D^3}}$ with $q_{sv}$ $[m^3/s/m]$ the unit solid volume transport: $q_{sv} = Q_{sv}/W$ with $Q_{sv}$ $[m^3/s]$ the solid volume flow rate, $W$ $[m]$ the river width, $\rho_s$ $[kg/m^3]$ the density of the sediments, $\rho$ $[kg/m^3]$ the density of water, $g$ the gravity acceleration and $D$ $[m]$ the grain diameter. $K/K'$ is the ratio between the flow Strickler coefficient $K$ and the grain roughness coefficient $K'$. This term makes it possible to correct the total constraint in order to take into account only the grain shear stress. $K$ is given by $K = \frac{U}{S^{1/2}R^{2/3}}$ and according to Meyer-Peter and Müller

(1948) the grain roughness coefficient can be estimated as a function of grain size distribution $K' = \frac{1}{n} = \frac{26}{D_{90}^{1/6}}$, with $D_{90}$ the diameter at about 90% by weight of the grains [m]. $\tau *$ $[-]$ is the Shield number, calculated as $\tau* = \frac{\tau}{g(\rho_s - \rho)D}$ with $\tau$ $[N/m^2]$ the shear stress.

This formulation is primarily based of laboratory experimentation with uniform and non-uniform sediments. It is one of the most used formulas when it comes to studying a river or a laboratory case study with a heterogeneous grain-size. This

characteristic makes it adapted to the LDG reach. However, the fact that it is only calibrated on laboratory measures can lead to non realistic results with in-situ input data. Besides, the Meyer-Peter-Müller equation is an excess shear relationship and its original formulation considers a critical Shields parameter equal to 0.047 as a threshold for characterizing the incipient motion of bed grains.

The Recking formula:

This non-threshold formula results from the work of Recking (2010, 2013b) and Recking et al. (2016). We used the version of this formula compatible with 2D calculation and local data (Recking et al., 2016). It can be written as follows (Eq. 6):

$$q_b^* = \frac{q_b}{\rho_s \sqrt{g(s-1)D_{84}^3}} = 14 \frac{\tau^{*\,2.5}}{1 + \left(\frac{\tau_m^*}{\tau_{84}^*}\right)^{10}} \tag{6}$$

$q_b^*$ $[-]$ is a dimensionless bedload discharge, $q_b$ $[kg s^{-1} m^{-1}]$ is the unit bedload discharge per unit width, $s = \rho_s/\rho$ is the specific gravity, and $g$ the gravity acceleration. $\tau_{84}^*$ $[-]$ is the Shield number, calculated from the diameter $D_{84}$: $\tau_{84}^* = \frac{\tau}{g(\rho_s - \rho)D_{84}}$ with $\tau$ $[N/m^2]$ the shear stress. Here the calculations were made using $D_{84}$ as the grain diameter. The parameter $\tau_m^*$ is a mobility term that defines the transition between partial transport $(\tau^* < \tau_m^*)$ and full mobility $(\tau^* > \tau_m^*)$ (Recking et al., 2016). The Recking formula was calibrated on field data $(\tau^* < \tau_m^*)$ and laboratory data $(\tau^* > \tau_m^*)$. It is the value of $\tau_m^*$ that gives its shape to the model. Therefore the value of $\tau_m^*$ strongly impacts the result, and its determination is difficult, especially for mountain streams. Ideally it should be based on measurements. Failing that, the available data suggest that an estimate is possible using Eq. 7 (Recking et al., 2016).

$$\tau_m^* = 0.26 S^{0.3} \tag{7}$$

The parameter $\tau_m^*$ is a mobility term that defines the transition between partial transport $(\tau^* < \tau_m^*)$ and full mobility $(\tau^* > \tau_m^*)$ (Recking et al., 2016). The Recking formula was calibrated on field data $(\tau^* < \tau_m^*)$ and laboratory data $(\tau^* > \tau_m^*)$. It is the value of $\tau_m^*$ that gives its shape to the model. Therefore the value of $\tau_m^*$ strongly impacts the result, and its determination is difficult, especially for mountain streams. Ideally it should be based on measurements. Failing that, the available data suggest that an estimate is possible (Recking et al., 2016). The Recking formula was coded in the subroutine "qsform.f" as it was not available among the proposed sediment transport equation in the SISYPHE module.

The main advantages of this formulation are that (Gonzales De Linares et al., 2020):

- it considers partial transport ;

- it has been developed based on field data, which makes it adapted to cross-section averaged calculations ;

- it has been validated with a wide data set for different independent watercourses ;

- it is adapted to mountain and Piedmont rivers with steep slopes and coarse grain-size.

## 4   Method

A model was developed at the LDG's reach scale to reproduce the hydrodynamic and morphodynamic processes that occurred during the 10-year return flood of June 2018. In fact, it was the only event for which we had the before and after topo-bathymetric data, necessary to check the model's ability to reproduce the observed morphological modifications. The followed methodology considered field data collection for the model's development and performance evaluation, the model generation, the selection of a relevant hydrodynamic model, after which a clear hydrodynamic calibration with a fixed bed to select the riverbed roughness was performed, to finally run the morphodynamic model with the two different bedload transport formulas.

## 4.1 Input data

### 4.1.1 Field data

The model starts at the junction of the "Gave de Gavarnie" and the "Gave de Cauterets" and extends up to the weir of the municipality of "Agos-Vidalos" (Fig. 5).

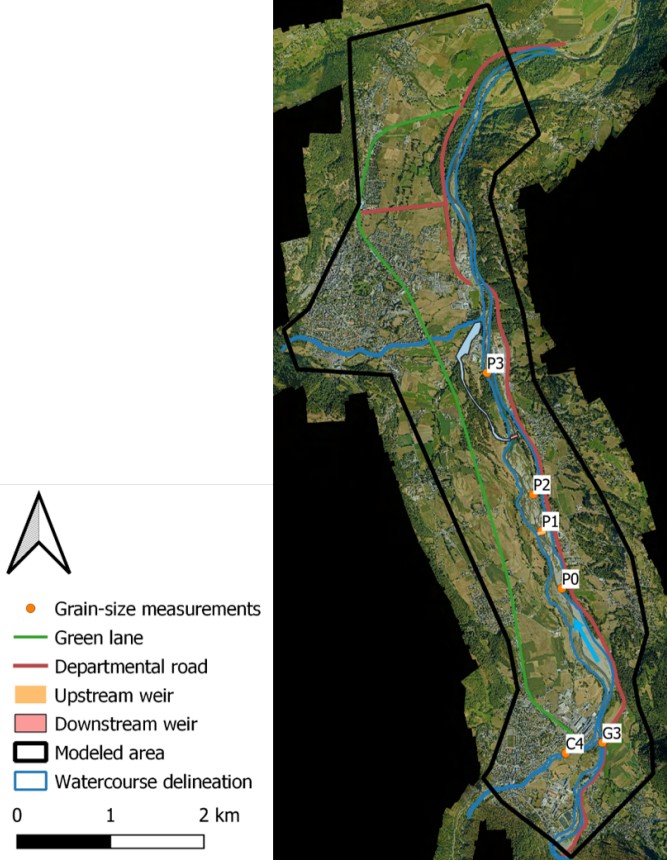

**Figure 5.** Overview of the considered area for hydromorphological modeling and identification of the different areas of interest

The available field data for the model's implementation are:

- a LiDAR DEM surveyed in 2016. The planimetric resolution is 1 meter and the Z precision is 1 centimeter;

- a LiDAR DEM surveyed in 2019 a few months after the flood of June 2018 ;

- dredging data (SHEM) provided by the former operators of the weirs. This data gives information on the possible bedload fraction that fills the LDG. Unfortunately, no grain-size distribution was available;

- grain-size data, collected on the ground over four sediment bars along the considered river reach. The hydromorphody-namic computations considered only the $D_{50} = 50\ mm$ for the MPM formula and the $D_{84} = 163\ mm$ for the Recking formula. These data were collected thanks to the Wolman sampling technique (Wolman, 1954), upstream the Beaucens weir. C4 and G3 are the grain size distributions on the Gave de Cauterets and the Gave de Gavarnie, upstream tributaries of the Gave de Pau;

- hydraulic data, representing water levels surveyed during the recession time of the 2018 flood event.

It is common to use sediment traps dredging data to estimate event-driven sediment transport in mountainous catchments as its measurement can be complicated in such flow conditions (Liébault et al., 2010). It appears that the LDG seems to have similar behaviour, even if it was not designed for this purpose. However, the recorded volumes represent both very fine sediments probably transported by suspension and very coarse sediment via bedload transport. To discriminate bedload from suspended load, coarse sediment dredging data over 11 years were collected upstream the first weir by the former hydro-power operators. The bedload volume is thus estimated to represent between 8 and 16% of the total transport, which is coherent with the feedback from the literature on similar configurations (Misset et al., 2020). This range of variation will be considered to compare simulated deposited volumes to observed ones.

### 4.1.2 Input hydrograph

The input discharges were generated by the physically based distributed hydrological model MARINE (Roux et al., 2011; Douinot et al., 2018; Roux et al., 2020) developed at the catchment scale. The data used for implementing MARINE model include rainfall (source: Météo France), topography (source: IGN), soil properties (source: INRA), land use (source: CORINE Land Cover)and event discharge (source: HydroEau France (DREAL) and EDF). The model is structured in three main modules. The first module separates precipitation into surface runoff and infiltration; the second represents subsurface runoff, and the last one represents surface runoff on slopes and in the drainage network. This last module is based on a transfer function that allows the routing of excess precipitation to the watershed outlet through the use of the kinematic wave approximation of the Barré-de-Saint-Venant equations. The spatial discretization of the catchment area is done using the grid resolution of the DEM.

The MARINE model is capable of simulating flood hydrographs at any point in the drainage network which is a real advantage in order to have an accurate approximation of the inputs to the Lac des Gaves system. Thus, three hydrographs were extracted for the big mesh (Fig. 6, left) for the three tributaries (the Gave de Cauterets, the Gave de Gavarnie and the Gave d'Azun) and one for the smaller mesh (Fig. 6, right). The details about the two different meshes are presented in the following section. This model has been calibrated based on the available observed discharges at three stations: the Gave de Cauterets, the Gave de Gavarnie and the Gave de Pau after the confluence with the Gave d'Azun. Six events extracted from these observed time-series allowed calibrating the model with a good confidence.

 ## 4.2 Model setup

### 4.2.1 Mesh generation

We built unstructured triangulated meshes using the software BlueKenue[2]. Sediment transport modelling is very sensitive to mesh size. Thus, two approaches were considered to create the meshes (Fig. 6):

– an eight kilometers long unstructured triangulated mesh that covers the whole study area (355 062 elements) was built.
     The mesh size is 3 m within the watercourse, 2 m in the fishery water intake area and 100 m in the floodplain ;

– a finer two kilometers long mesh in the LDG area around the two weirs (201 569 elements). The mesh size for this smaller domain is 1 m in the riverbed, 2 m in the fishery water intake and 20 m in the floodplain.

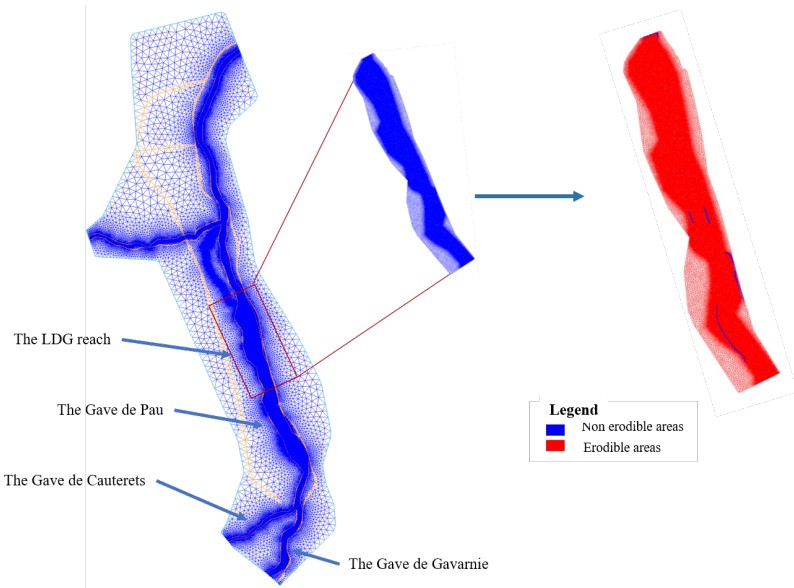

**Figure 6.** Considered meshes for the hydromorphological modelling. The orange lines represent soft-lines corresponding to roads or river banks where we force the mesher to pass through

The finer mesh covers a much smaller area, so it is used to perform a less time consuming fine analysis of the sediment transport behaviour around the area of interest: the LDG between the two weirs. The obtained results with this small mesh  allowed us to pick the best performing parameters for the whole domain with which we only simulated restoration scenarios, resulting in a substantial saving of time. To represent the anthropogenic structures along the river, fixed embankments, weirs and rip-raps, were considered as non erodible (blue in Fig. 6) in the context of sediment transport computations. The simulations with both meshes considered sediment transport.

---

[2]http ://www.nrc-cnrc.gc.ca/fra/solutions/consultatifs/blue kenue index.html

#### 4.2.2 Boundary conditions

For the mesh representing the entire study area, four boundary conditions were defined. Upstream, discharges are set as an input for the "Gave de Gavarnie", "Gave de Cauterets" branches and the "Gave d'Azun" branch downstream the LDG. The downstream boundary condition is a free surface elevation determined by a rating curve calculated with a weir law (Eq. 8) respecting the characteristic of the "Agos-Vidalos" weir:

$$Q = L \times \mu \times \sqrt{2g} \times (h - Z_{weir})^{(3/2)} \tag{8}$$

with $L$ [m] the spillway width, $g$ [$m.s^{-2}$] the gravitation acceleration, $h$ [m asl] the free surface elevation, and $Z_{weir}$ [m asl] the weir elevation.

At this location, the bed slope is 0.018 m/m (Fig. 6). The particularity of this boundary condition is that it delivers sufficient bedload at the model inlet to keep the riverbed elevation at the inlet cross-section constant in time. It has been assumed that the upstream boundary condition on solid discharge has low influence in the area of interest which is the Lac des Gaves: the upstream condition is located sufficiently far from it to reduce its influence. This is a relatively good assumption for the flood event of 2018 for which little material seems to have come from upstream the area of interest.

As for the sediment transport boundary conditions, we first attempted to prescribe solid discharges estimated thanks to the 2018 hydrograph and the Recking (2013a) formula as no bedload measurements were available for this event. Unfortunately, this generated many instabilities around the upstream boundary that led to aberrant erosions or sediment depositions, extremely high and localized on only one or two cells around the upstream boundary. To overcome this limitation, a morphological equilibrium condition is set at the inlet (Tassi and Villaret, 2014). The particularity of this boundary condition is that it delivers sufficient bedload at the model inlet to keep the riverbed elevation at the inlet cross-section constant in time.

For the smaller mesh, two boundary conditions were defined. The 2018 flow hydrograph for the "Gave de Pau" river is set as an input upstream. The downstream boundary condition is a free surface elevation estimated with the same weir law presented above (Eq. 8) respecting this time the characteristic of the "Préchac" weir.

### 4.3 Calibration strategy

#### 4.3.1 Hydrodynamic model

In a classical way, hydrodynamic calculations are first carried out. To calibrate the hydrodynamic, simulations on a steady state were performed for a discharge of 58.4 $m^3/s$ measured on July 9, 2018 measured by a public service in France named DREAL and represented in Fig. 7. Under the hydrodynamic calibration conditions, morphological changes and bedload transport were limited. The fact that no bathymetric data was available can lead to a non-negligeable uncertainty on the water surface elevation. However since the water depth during the LiDAR surveys was approximately the same we can consider that this is acceptable compared to other uncertainties.

In the TELEMAC-MASCARET system, two categories of parameters can be adjusted: the numerical parameters (time step, type of solver and its accuracy) and the physical ones (De Linares, 2007). In our case, we focused on the physical one with a

variation of Strickler friction coefficient from 20 $m^{1/3}/s$ to 60 $m^{1/3}/s$. Water surface measurements along the river are used to quantify the simulations' accuracy. Model's accuracy is evaluated using the Root Mean Square Error (RMSE).

The best results were obtained with the $K = 30 \, m^{1/3}/s$ friction coefficient for the constant viscosity turbulence model (Fig. 7). The LDG area is well represented as the errors do not exceed 20 $cm$ between the two weirs and the RMSE value obtained for this simulation is 0.31. The 1D longitudinal profiles presented in this paper are plotted from an extraction of the lowest bathymetric points of the 2D model.

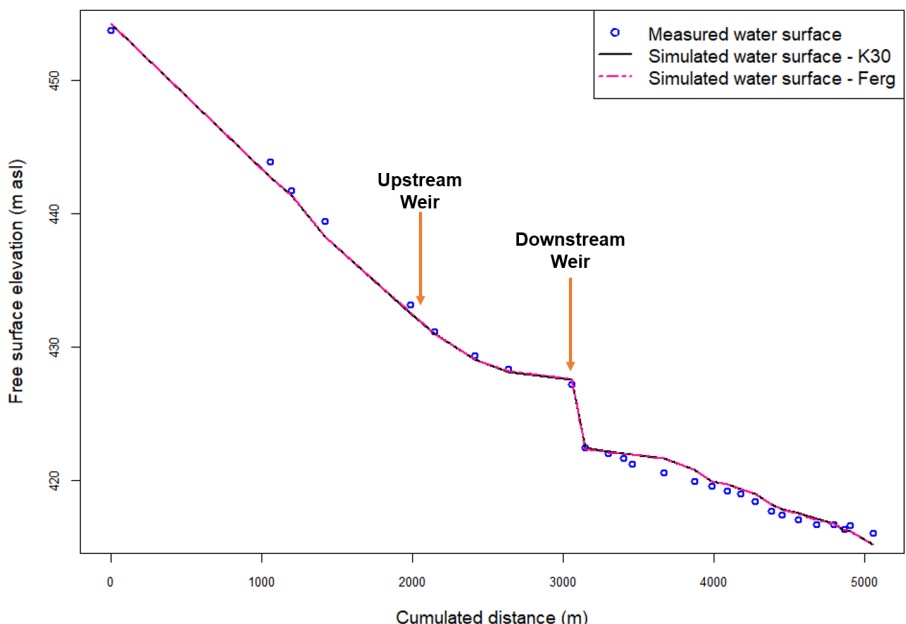

**Figure 7.** Longitudinal profile of observed and simulated water surfaces corresponding to a discharge of 58.4 $m^3/s$ measured the 09/07/2018 - K30 (resp. Ferg): simulation with Strickler (resp. Ferguson) friction law

### 4.3.2 Hydro-morphodynamic model

The hydro-morphodynamic simulations were based on the flood event of 2018 that we assumed responsible for the visible morphological changes between the two topographic campaigns of 2016 and 2019. Unfortunately, there have been no topographic campaigns between 2016 and 2019 that would account for the effects of the 2018 flood only.

First, the friction coefficient and the turbulence model selected during the hydrodynamic calibration process were used for the first hydro-morphodynamic simulations. Then other simulations for the two sediment transport (MPM and Recking) and friction formulas (Ferguson and Strickler) were performed. The specific parameters of each sediment transport formula

(Shields number, MPM coefficient, slope effect, etc.) were tested afterwards to analyse their influence on the performance of the simulations.

## 4.4 Performance evaluation

### 4.4.1 General comparison of erosion and deposition areas

The model's qualitative performance evaluation was first done by visually comparing the simulated geomorphological evolution maps to the DEM of difference (DoD). This allows a qualitative evaluation of the model's capacity to reproduce the spatial variability of the processes (erosion and deposition) and to locate possible aberrations.

Longitudinal profiles and cross-sections where significant morphodynamic processes occurred were also compared to acquire a more refined vision of the evolution at the local scale. However, in braided rivers, as in the "Gave de Pau", a significant variability is observed between two measurements knowing channel migration phenomena.

### 4.4.2 The Brier Skill Score

The model's calibration requires a considerable amount of computations where different parameters are modified. Identifying rapidly the best performing with a cost function can thus be time saving. The selected cost function to do this is the Brier Skill Score (BSS). It was developed initially for the assessment of meteorological model's performance and uses a baseline prediction to quantify the model's new prediction skill. Furthermore, during the last decades, many hydromorphological studies considered it to evaluate the model's skill to simulate the sediment erosion and deposition processes along the whole domain (Aguirre et al., 2020; De Linares, 2007; Sutherland et al., 2004). It can be expressed as follows (Eq. 9):

$$BSS = 1 - \frac{\frac{1}{N}\sum_i^N (y_i - x_i)^2}{\frac{1}{N}\sum_i^N (b_i - x_i)^2} \tag{9}$$

with $N$ the number of measurement points, $b_i$ baseline, here we use initial river bed elevation (DEM of 2016), $x_i$ observed river bed elevation (DEM of 2019) and $y_i$ the simulated river bed elevation. Table 2 shows the recommended model performance classification for the BSS.

One of the major advantages of considering the BSS for the hydromorphological model's performance is that its value is not impacted if measurement points did not present any evolution. The number of considered grid points should only have little influence on the BSS value.

### 4.4.3 Comparison of the deposited volumes

Using a cost function to evaluate a hydromorphological model's performance with a braided morphology can be quite pessimistic. To date, numerical models cannot predict channel migration processes that occur in braided rivers. These phenomena are uncertain and random. A modeler should thus not expect the model to predict channel migration accurately during a flood. Despite these limitations, the choice of a 2D model has been made because it allows a better representation of the hydrodynamics and in particular of the friction with taking into account a spatialization of the water height. Even if the representation of

**Table 2.** Classification of BSS values for model performance evaluation (Aguirre et al., 2020)

| Performance of the simulation | BSS Value |
|---|---|
| Excellent | 1.0 – 0.5 |
| Good | 0.5 – 0.2 |
| Reasonable | 0.2 – 0.1 |
| Poor | 0.1 – 0.0 |
| Bad | < 0.0 |

the braiding and of the different flow arms is not the real one, the 2D model has the advantage of a continuity of the dynamics,
contrary to the 1D model with interpolation between two profiles and water height projected on the DEM to estimate the extent
of the flooded area.

As the issue here is the filling of the LDG and the high amount of sediments that might be delivered to the downstream system
if the weirs are levelled, the comparison of the simulated deposited volumes with the field data appears to be a relevant model's
performance indicator. Field erosion and deposition areas were estimated through topo-bathymetric differencing between two
LiDAR DEMs surveyed in 2016 and 2019 (Fig. 8). The 10-year return period flood of June 2018 is considered the only
morphodynamic flood that occurred during this period.

### 4.4.4 Statistical distribution of erosion and deposition

We also considered plotting on a histogram the area of bed experiencing morphological changes as performed by Williams
et al. (2016b). The statistical distribution of erosion and deposition could then be qualitatively compared; this was particularly
useful for comparing simulations with different bedload transport and friction equations.

## 5 Results

The TELEMAC-MASCARET modelling system can run in parallel mode using domain decomposition and MPI based codes.
The calibration scenarios have been carried out on a Linux server over 16 processors at the Institut de Mécanique des Fluides
de Toulouse (IMFT).

The performance of the two friction laws and the two bedload formulas was assessed with the three performance indicators:
longitudinal profiles and cross-sections comparison, BSS scores along with the whole domain and the comparison of the
deposited volumes is also considered.

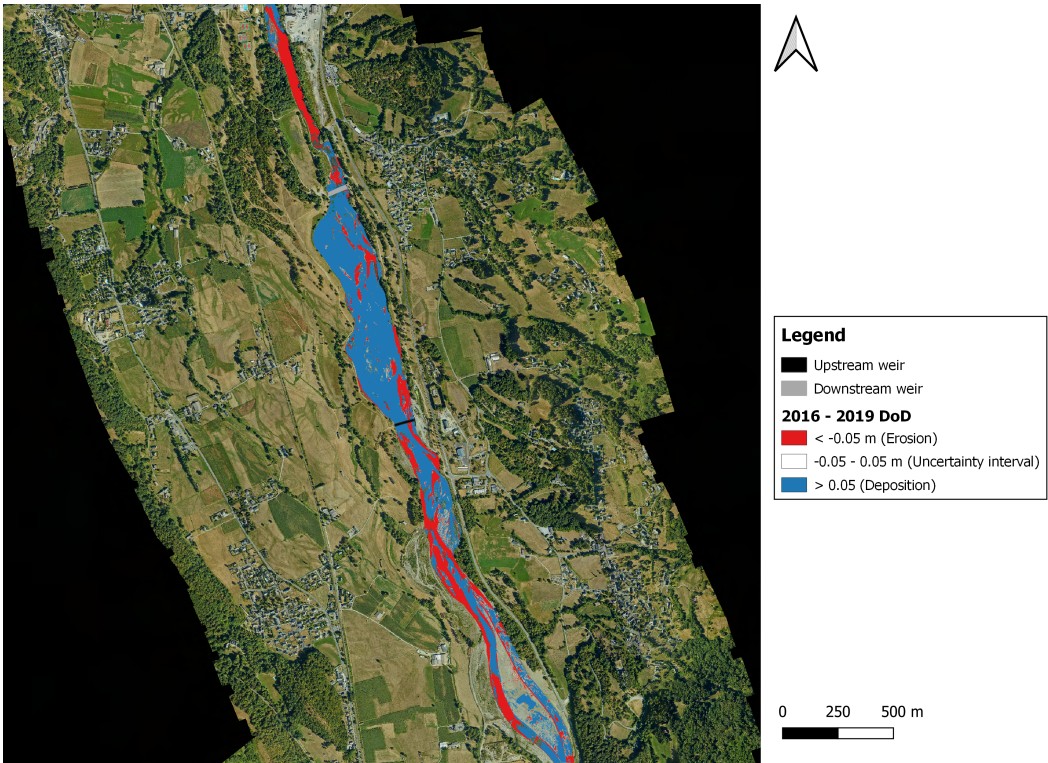

**Figure 8.** Eroded (red) and deposited (blue) areas in the LDG reach estimated through topo-bathymetric differencing between the two LiDAR DEMs surveyed in 2016 and 019. The upstream weir is represented in black and the downstream weir in grey. This figure illustrates the filling of the LDG as almost all its surface represents deposited materials

## 5.1    General visual comparison of eroded and deposited areas

The simulated results are compared only within the intersection of areas that were emerged during the two LiDAR campaigns (2016-2019) as this technique does not collect submerged bathymetric data. In general, the model seems to correctly represent the filling tendency of the LDG as a deposition front can clearly be observed, which is coherent with what is observed in the field (Fig. 9). The simulated evolution shows that the selection of a friction law has more impact on the results than the sediment transport equation. In fact, for both sediment transport formulas, the Strickler friction law tends to overestimate erosion and deposition processes within the LDG reach, whereas they seem to be minimised with the Ferguson friction law (Fig. 9 and 10). The observed evolution (Fig. 9 a) seems to be more matching with the results obtained with the MPM bedload transport formula and the Strickler friction law (Fig. 9 d). This is probably due to the fact that the Strickler formula seems to overestimation erosion and deposition processes. Since the simulations only take into account bedload and the difference in DEMs obviously represents total load, it is therefore logical that the results of the formula overestimating the bedload seem

closer to the observations of the total load. Of course this is only a qualitative way of comparison. The volume of erosion and deposition by bedload only for all these cases are compared later in the article (§5.3, Table 3).

If we compare the obtained results with the Ferguson friction law for the two bedload transport formulas, the simulations with the MPM formula tend to predict higher sediment deposition and erosion amounts. As it is a threshold formula, the results below or around the critical shear stress can be poor because of a zero prediction or an overestimation of sediment transport. This formula is considered efficient when $\tau_{84}/\tau_{c84} > 2$ and thus for rather strong flow conditions (Recking et al., 2012). In our case, as we consider the whole 2018 hydrograph (10 days), we are not always in these conditions. Hence, the observed over-estimations can be due to fluctuations around the threshold during the 2018 flood event.

The morphodynamics around the upstream weir (Fig. 10) are more complex. During the 2018 flood event, two main river branches were created (left and right bank) with very strong erosions around the right bank and considerable depositions elsewhere. The comparison with the experimental evolution has to be taken carefully as the bathymetry under the water surface cannot be captured by the LiDAR technique. However, as the two measurement campaigns were conducted in severe low-water seasons, we can thus estimate that these uncertainties are reduced. Figure 10 shows that the two sediment transport formulas can reproduce the observed erosion processes around the right bank but to a lesser extent. The strong erosion observed in the right bank during the flood (pink in Fig. 10a) is actually due to the disappearance of a river protection that has been washed out by the flood. It seems like the model is not able to reproduce this phenomenon. In fact, the maximum simulated water extent for both sediment transport formulas (Fig. 10b and 10c) does not reach this area which explains why this extreme erosion process is not reproduced. However, the MPM sediment transport formula tends to estimate stronger erosion and deposition processes compared to the Recking formula that seems to minimise them.

## 5.2 Longitudinal profiles and cross-section comparison

Further quantitative investigations were done by comparing longitudinal profiles for both sediment transport formulas and friction laws. For the MPM formula, longitudinal profiles comparison confirms that the Strickler friction law tends to overestimate bedload depositions within the LDG (Fig. 11). The same conclusions can be drawn with the Recking formula and Strickler friction law (Fig. 12). It seems thus that the Ferguson law is the one providing the most realistic results for both bedload formulas, which is coherent with the visual interpretation made in section 5.1.

The longitudinal profiles comparison for both bedload transport formulas with the Ferguson friction law (Fig. 13) shows that MPM formula tends to overestimate sediment depositions within the first half of the LDG (950-1400 m) compared to the Recking formula. However, the sediment deposition front progression seems to be more accurately simulated with the MPM formula. Visually speaking, the simulated long profile with the Recking formula seems to be closer to the observed evolution. This is confirmed by the score of this simulation (BSS = -0.01) calculated over the long profile, whereas the MPM formula has a score of -0.3. As we can see, both simulations seem to perform poorly according to the BSS criteria only. This questions the relevance of this criteria for complex morphologies such as the braided LDG reach as the hydromorphological model cannot simulate exactly where a channel or a bar will be.

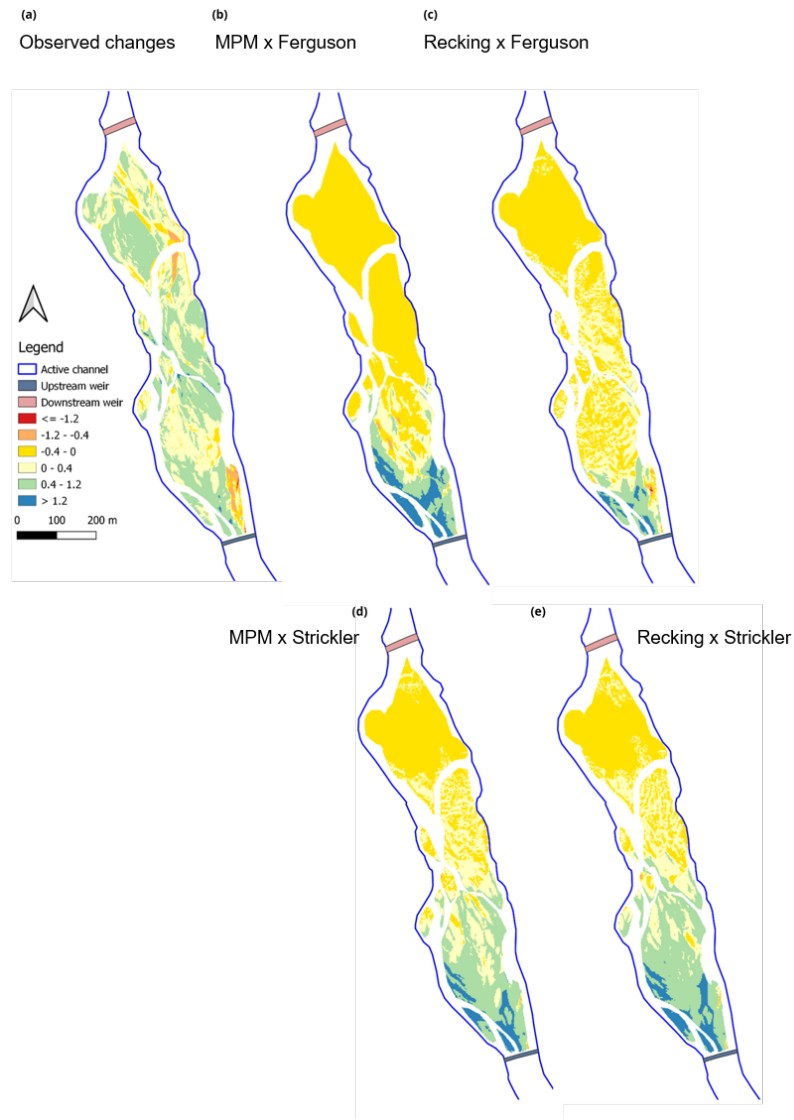

**Figure 9.** Comparison of the simulated evolution with the two bedload formulas (MPM and Recking) and the two friction laws (Strickler and Ferguson). (***a***) is 2016-2019 DoD; the simulated evolutions are represented with (***b***) the MPM formula and the Ferguson friction law; (***c***) the Recking Formula and the Ferguson friction law; (***d***) the MPM formula with the Strickler friction law and (***e***) the Recking formula with the Strickler friction law.

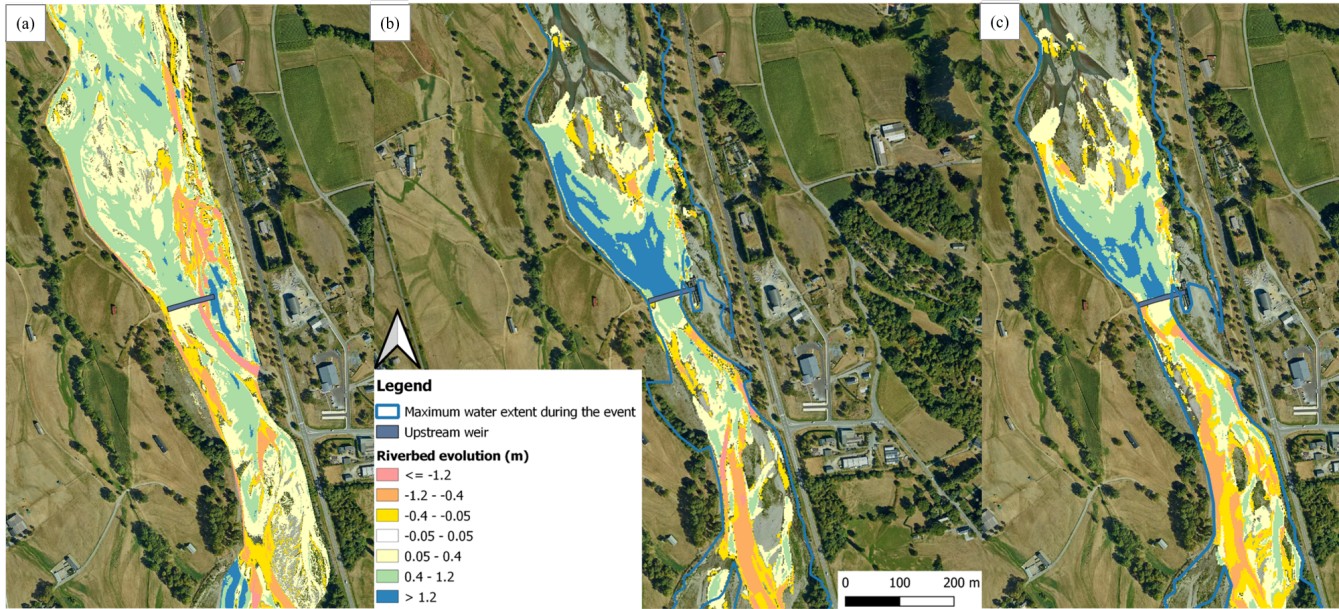

**Figure 10.** Channel evolution around the Beaucens weir. (***a***) is the experimental evolution estimated through the difference between the 2016 and the 2019 topography; (***b***) is the evolution estimated with the MPM sediment transport equation and the Ferguson friction law and (***c***) is the simulated evolution with the Recking sediment transport formula and the Ferguson friction law.

The evaluation of the model's performance over the cross sections confirms this statement (Fig. 14). As mentioned above, Figure 14 shows that the model experiences difficulties to geographically match the location of channels and bars forming the braided morphology of the LDG reach. Strong erosions are observed for both formulas upstream the Beaucens weir within
465 the main channel, which is not coherent with the observations on the ground. However, this comparison can be biased by the fact that we are manipulating LiDAR data, not taking into account the bathymetry below the water surface. For the upstream cross-section the two bedload transport formula perform poorly with a BSS of -0.04 for the MPM formula and -0.06 for the Recking formula. The comparison of the cross-sections downstream the weir shows that the Recking formula seems to have a better performance (BSS = 0.35) compared to the MPM sediment transport formula (BSS = -1.2) that overestimates sediment
depositions within the lake.

Finally, this topographic examination questions the classical performance analysis methods for morphodynamic models. Knowing the multiple variabilities in a mountain braided watercourse, performance criteria combined with local altimetric analysis might be too strict and incomplete to assess the ability of the model to reproduce the mobilised sediment volumes over a flood event. As the aim is to give the local elected representatives indications regarding possible sustainable restoration
scenarios, a volumetric analysis can provide valuable additional insights as it gives information on the possible volumes that might end up downstream if a weir lowering/removal solution is considered.

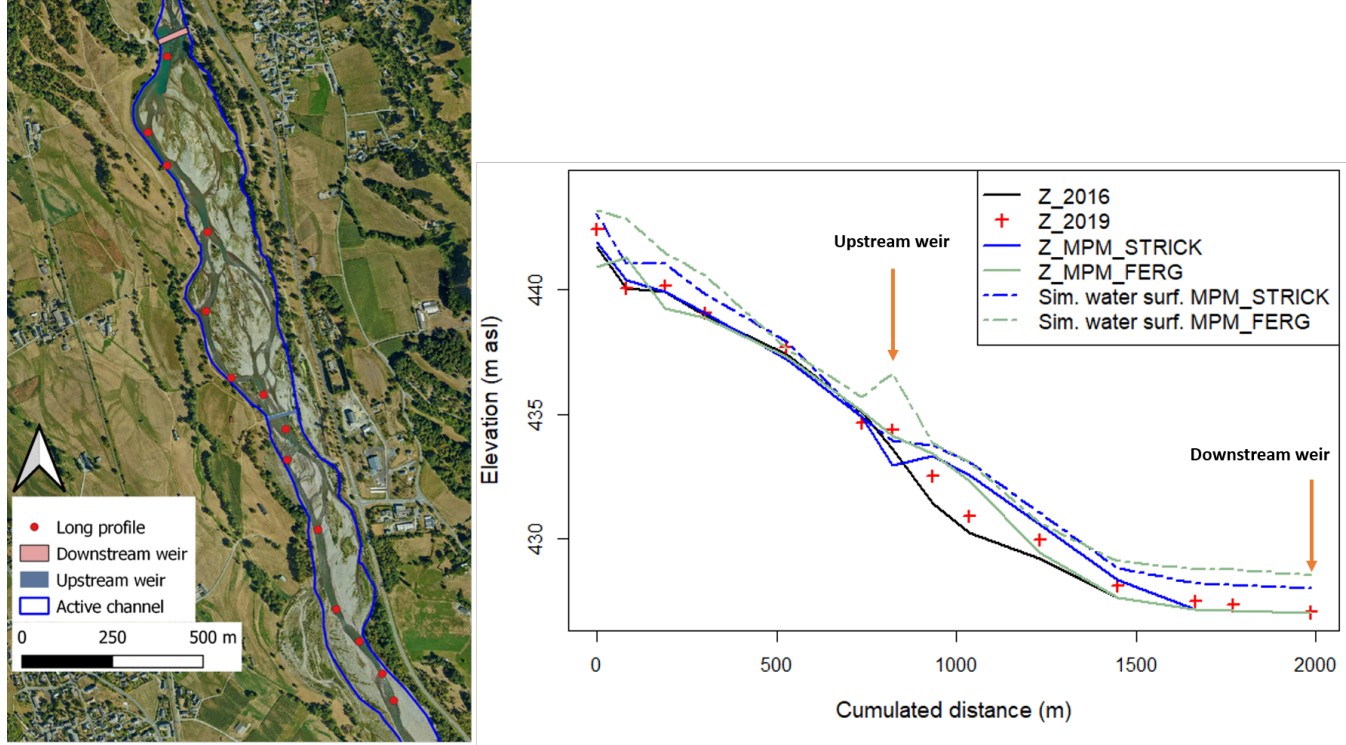

**Figure 11.** Simulated longitudinal profiles (solid lines) and maximum simulated water surface (dashed lines) comparison for the MPM formula and the two considered friction laws: STRICK for Strickler friction law, FERG for the Ferguson friction formula - Z_2016: DEM of 2016, Z_2019: DEM of 2019. The first point of the long profile is not located at the inlet of the model

### 5.3   Comparison of deposited volumes

To compare the simulated deposited volumes to the ones observed in the field, the score $r = V_{scal}/V_{smeas}$, with $V_{scal}$ $[m^3]$ the simulated deposited volume and $V_{smeas}$ $[m^3]$ the measured deposited volume, was calculated between the two weirs (Table 3). When $r > 1$, this means that we overestimate sediment depositions within the LDG, when $r < 1$ it is the opposite: we tend to underestimate sediment deposition. The deposition phenomenon within the LDG is one of the most important processes to reproduce as it represents the potential volumes that might be mobilised if the weir lowering/removal restoration measure is considered.

The reconstruction of the filling of the lake through different periods has allowed the collection of interesting data that provides annual trends of material input. These results are derived from an analysis of bathymetric profiles from which the volumes were extracted. For the flood of 2018, a total (bedload and suspension) sediment deposition volume of $81220$ $m^3$ was estimated. In order to distinguish the bedload phenomenon and as our model only considers this process, dredging data were collected for approximately ten years. These data allowed us to estimate the fraction of bedload of the total sediment transport.

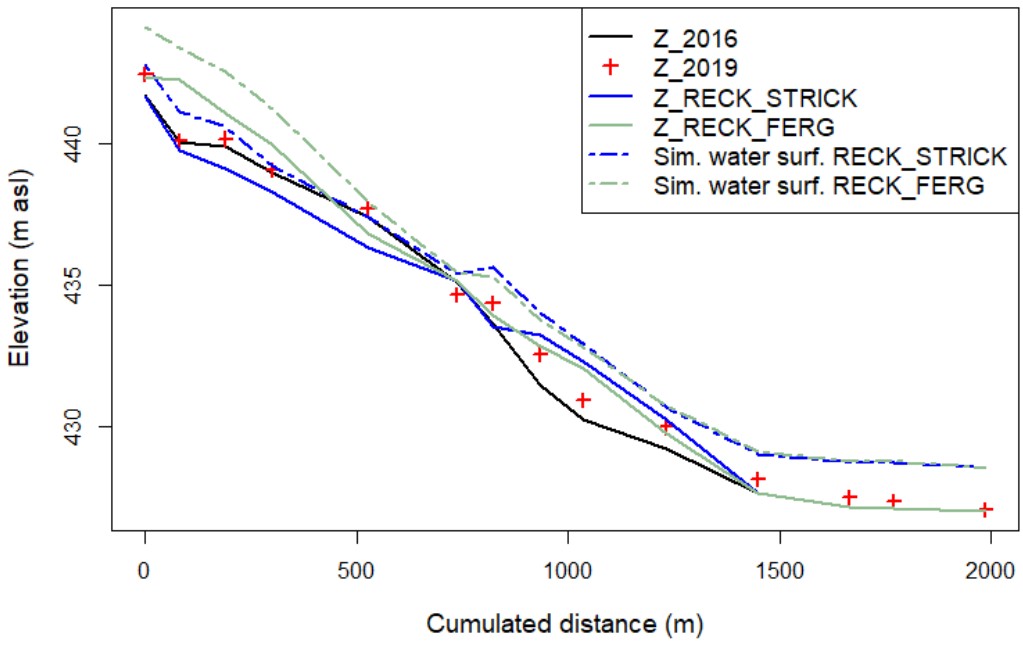

**Figure 12.** Simulated longitudinal profiles (solid lines) and maximum simulated water surface (dashed lines) comparison for the Recking formula and the two considered friction laws: STRICK for Strickler friction law, FERG for the Ferguson friction formula - Z_2016: DEM of 2016, Z_2019: DEM of 2019

This fraction is estimated to be between 8 and 16% of the total transport, representing the lower and upper uncertainty interval limits.

As mentioned above, the model only considers bedload transport. Hence, the score calculation was performed on the fraction of sediments estimated to be deposited via a bedload transport process. Generally speaking, the model seems to simulate sediment depositions close to the upper interval limit (16% of the total deposited volume observed including both suspension and bedload transport). The best results ($r = 1.53$) seem to be obtained with the combination of the Recking bedload transport formula and the Ferguson friction law, which is coherent with the interpretations made above with the longitudinal profiles comparison. The MPM formula tends to overestimate the deposited volumes; however, acceptable results were obtained with the Ferguson friction law ($r = 2.44$). This supports the fact that Ferguson friction formula seems to be the most suitable one for the LDG reach.

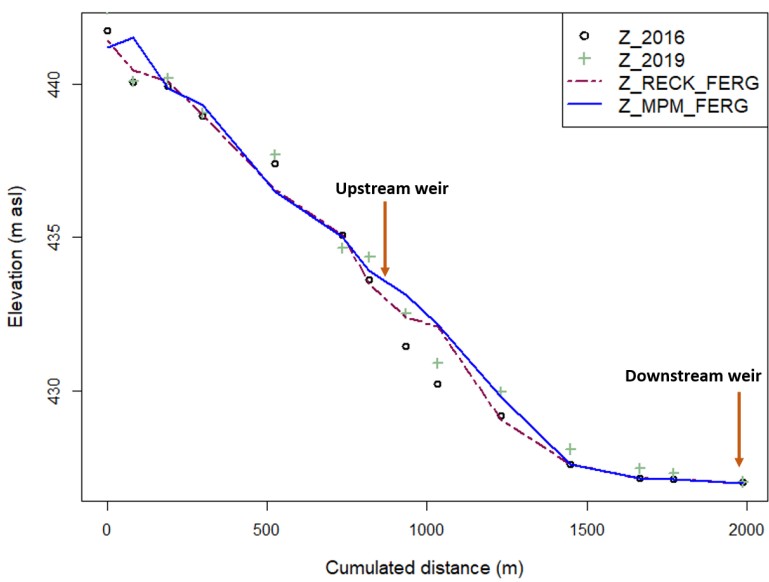

**Figure 13.** Simulated longitudinal profiles comparison for the Recking and the MPM bedload formulas and Ferguson friction law - Z_2016: DEM of 2016, Z_2019: DEM of 2019

**Table 3.** Comparison of simulated deposited volumes and observed ones using score $r = V_{scal}/V_{smeas}$. The lower interval limit represents 8% of the total sediment transport and the upper interval limit 16% of the total transport

| | Total deposited volume (m³) | Simulated deposition score | Simulated bedload volume score *Lower Limit* | Simulated bedload volume score *Upper Limit* |
|---|---|---|---|---|
| MPM x STRICK | 88 362 | 1.09 | 13.6 | 6.80 |
| MPM x FERG | 31 761 | 0.39 | 4.89 | 2.44 |
| RECK x STRICK | 58 354 | 0.71 | 8.99 | 4.49 |
| RECK x FERG | 19 885 | 0.24 | 3.06 | 1.53 |

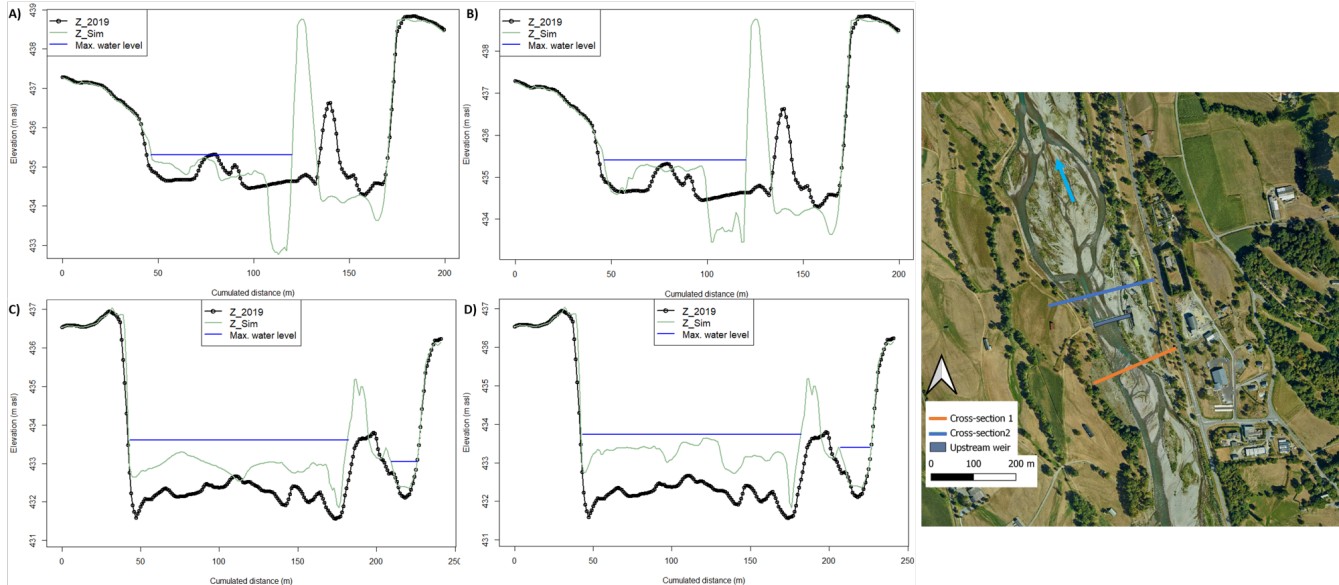

**Figure 14.** Cross-section comparison upstream (orange cross-section) and downstream (green cross-section) the Beaucens weir (right). The presented simulations were performed with the Ferguson friction law. Left: (**a**) and (**b**) are the usptream cross-sections for the Recking formula and the MPM formula respectively ; (**c**) and (**d**) are the downstream cross-sections for the Recking formula and the MPM formula respectively

## 5.4    Distribution of erosion and deposition

The statistical distribution was analysed upstream the LDG (Fig. 15 b)) and within the LDG (Fig. 15 a)). The simulated bed evolution for both bedload transport formulas and friction equations is lower than the observed bed evolution. It is able to better reproduce the spatial distribution of bed evolution even if a calibration process should be done to reproduce the real evolution. The distribution of bed elevation changes with the Ferguson friction equation corresponds more closely to the one observed. We also observe that the Strickler friction equation has a completely different distribution and is characterized by a 505    longer deposition tail attributed to more important sediment depositions upstream and within the LDG. As we can see, model predictions were relatively sensitive to the choice of friction equation and bedload transport formula.

## 5.5    Implementation of restoration scenarios

The model seems to provide reliable results with the Ferguson friction law, so this formula was selected to perform restoration scenarios simulations. The two bedload transport formulas (MPM and Recking) were considered. The MPM is considered 510    to provide more extreme results that we view as the "worst-case scenario", whereas the Recking formula is considered more realistic or even to minimise the transported volumes.

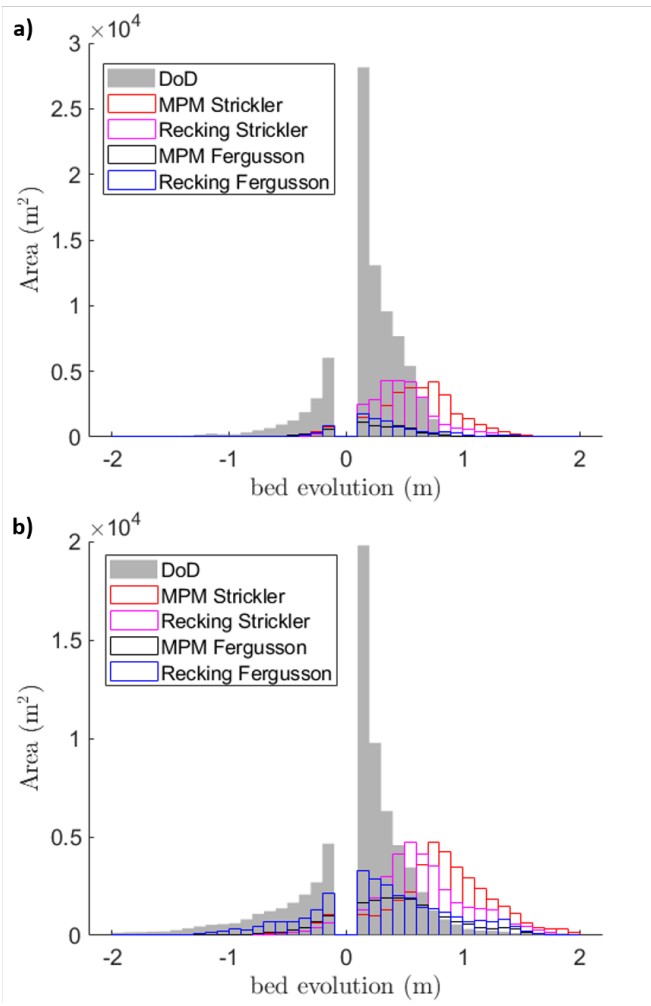

**Figure 15.** Statistical distribution of erosion/deposition. The grey histogram shows observed morphological change and the colored ones show model predictions with the different bedload and friction equations within the LDG a) and upstream the LDG b).

Two restoration scenarios were performed using the LiDAR DEM surveyed in 2019 as the initial topography. A Business as Usual Scenario (BAU) which corresponds to the current situation, where we consider that no restoration measure is implemented. A Weir Lowering scenario (WL), implemented through the modification of the bathymetry, for which the downstream

weir (Préchac) is lowered by two meters. To assess the influence of each scenario, two cross-sections upstream and downstream the weir were analysed (Fig. 16).

For each restoration scenario (BAU and WL), the first simulation considered the 2019 topography and a 2018 type flood event is injected. Then another 2018 type flood event is injected on the resulting bathymetries of these two first simulations (Z_Sim_BAU and Z_Sim_WL). As a result, we can have an overview of the LDG reach behaviour depending on the selected

restoration measure and the bedload formulas after two 10-year return period-like flood events. 10-year return period-like flood events were chosen because they are both relatively large flood events with a rather low return period. Besides, this was also discussed with the river managers, who wanted to be prepared for such kinds of events that might occur more and more frequently.

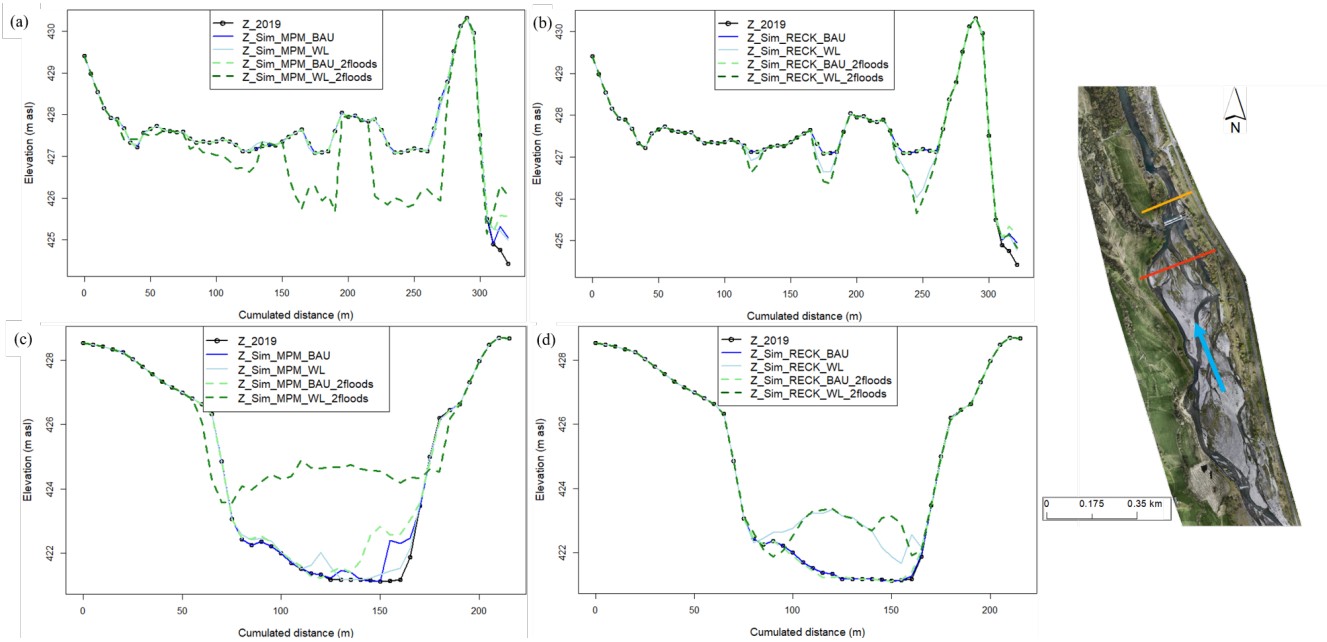

**Figure 16.** (*a*) and (*b*) are the simulated cross sections for the MPM formula and Recking formula respectively upstream the Préchac weir (red cross-section on the right) for the BAU and the WL scenarios. (*c*) and (*d*) are the simulated cross-sections for the MPM formula and the Recking formula respectively downstream the Préchac weir (orange cross-section on the right) for the same scenarios. Z_2019 represents the LiDAR DEM surveyed in 2019 that is the initial topography.

For the BAU scenario, very few morphological changes are observed upstream of the "Préchac" weir. Downstream, some

depositions are observed with the MPM formula for the two flood scenarios (1 and 2 consecutive 2018 flood events). This

consolidates the observations made above regarding the fact that this sediment transport formula tends to estimate more important sediment transport volumes than the Recking one for high flow situations. Very few changes are observed for the Recking formula for the BAU scenario.

As expected, the WL scenario shows more morphological modifications for both sediment transport formulas. Upstream the "Préchac" weir, very few changes are observed for the one 2018 flood event scenario with the MPM formula. However, severe incisions are noticed for the two 2018 flood events scenarios (up to - 2 meters). For the Recking formula, the incision phenomenon is less exacerbated for the two 2018 flood events scenario. The opposite phenomenon is observed downstream for both sediment transport formulas but with different amplitudes. Impressive sediment depositions can be noticed with the MPM formula for the two flood events scenario (up to + 2.5 meters). Considerable aggradation for the Recking formula can be observed as well but with a lesser amplitude (up to + 1 meter).

The surprising nature of the results is the fact that the reaction of the model with the MPM formula for the one 2018 flood event seems to be very modest, whether it is upstream or downstream the "Préchac" weir. This might be due to the fact that this is a threshold formula and that the critical shear stress might not be exceeded in this area to generate sediment transport and thus morphological modifications. Conversely, as the Recking formula is not a threshold one, partial transport is estimated even for small discharges which can explain the observed morphological changes.

## 6  Discussion

### 6.1  Performance of sediment transport and friction equations

The results highlight the importance of the friction law as it conditions the shear stress calculations' results and thus bedload transport. Friction laws are equations that usually link flow velocity to depth and roughness (Rickenmann and Recking, 2011). The Manning-Strickler formula is particularly suited for high submersion flows (Ferguson, 2007; Rickenmann and Recking, 2011) whereas the Ferguson friction law is known to have the best performances from low to high submergence (Rickenmann and Recking, 2011) which is probably more suited to our case study, for which the water height is of the same order of magnitude as the roughness at the beginning of the flood. Then the submergence becomes high during the peak flood. This explains why sediment depositions and erosions appear to be overestimated with the Strickler formula. The Ferguson friction law seems thus better suited to the complexity of the processes that occur within the LDG.

For the two sediment transport formulas, the model predicts that all the depositions only occur within the first half of the LDG. This can be due to the fact that many factors are not considered in the model such as the consideration of the whole grain-size distribution for bedload transport. Besides, the downstream part of the LDG is mainly composed of very fine sediments. The morphological evolution within this section is thus, for the most part, probably due to suspended load, not considered in our model. In addition, it is likely that the roughness parameters used by the two considered friction laws ($K$ or $D_{84}$) are not constant in both space and time. This can also explain some of the differences between the modeled and measured evolutions.

## 6.2 Is the BSS adapted to complex morphologies?

Each simulation was assessed with the BSS since this score was considered relevant for morphological changes evaluation by the recent literature. More than 60 simulations were performed with different model parameters combinations. The best BSS results were obtained for simulations with very little riverbed changes (BSS = 0.06) that have not been presented here. However, this simulation was not the best performing one in terms of deposited volumes and long profile evolution. For the two best-performing simulations according to the longitudinal profile analysis and the deposited volumes (RECK x FERG and MPM x FERG), the BSS results are poor (-0.04 and -0.12, respectively).

This raises the question of the relevance of the BSS criterion for morphologies as complex as the LDG reach. The fact that the model cannot accurately reproduce the different river branches due to the braiding phenomenon is approached in a very strict manner by the BSS. To our knowledge, models have difficulties predicting channel migration processes that occur in a braided river as this phenomenon is uncertain and random, especially during flood events. Besides, our model was not developed to reproduce braiding or deal with suspended sediment; however it is one of a few models available that are able to model morphodynamics (erosion and deposition) during large flood events. Here we use this well constrained example to assess its ability to reproduce volumes and cross-sections, and assess its suitability as a tool to inform policy makers. Although the BSS can be advantageous as it helps to rapidly evaluate the performances of numerous simulations, using it on braided and thus complex morphologies does not seem relevant in our study site. Therefore another criterion adapted to braided rivers has also been considered: the statistical distribution of erosion and deposition and is considered better suited for such kind of morphology (Fig. 15 (Williams et al., 2016b)).

## 6.3 Restoration of the LDG

Two restoration scenarios were performed around the LDG reach: a Weir Lowering scenario (WL) and a Business As Usual scenario (BAU). As expected, the WL scenario showed significant morphological modifications for both sediment transport formulas (MPM and Recking) whereas the BAU scenario predicted very few changes. In any case, even if the observed morphological evolution after the weir lowering will considerably enhance the ecological situation of the LDG reach by reactivating the sediment continuity allowing the circulation of anadromous fishes, this scenario might pose serious operational problems for the river managers. The upstream incision can, for example, induce:

- temporary bank erosions that can lead to the loss of portions of agricultural lands ;

- the propagation of the incision upstream until it meets a blocking point (the "Beaucens" weir and then the same problem will be observed) ;

- the lowering of the water table on which the farmers depend ;

- the disconnection of the fishery water intake.

The significant depositions observed downstream can increase flood risks knowing that it is already vulnerable as many stakes are located in this area. As we can see in Fig. 16, the simulated river bed with the MPM formula for the two 2018 flood events is very close to the left bank's altitude.

However, our model's observed over-depositions and incisions can be criticized. Indeed, only bedload was considered with coarse sediments and not the total sediment mixture. The suspended load is completely neglected. Still, a significant percentage of the LDG is composed of very fine sediments (silt, clay), especially downstream, which might be mobilized quickly after an action on the downstream weir. This means that a portion of the mobilized sediment will certainly be flushed far downstream and not have time to settle and induce the observed modelled morphological changes. Depositions would still be observed

but to a lesser extent. Besides, our model only considers one homogeneous grain size in the whole domain, which might also explain the over-depositions observed downstream. A perspective of improvement for the model can thus be to consider the whole grain size distribution spatially distributed over the studied domain to have a more realistic view of the impacts of any restoration measure. This can be done with the brand new sediment transport module "GAIA" developed to handle grain-size issues better.

To sum up, our model reproduces realistic tendencies but can still be improved to make better volumetric estimations. One recommendation to decision makers is to not only consider the downstream weir but to consider both weirs in the restoration project. Besides, in such kinds of complex morphologies, the main advice is to consider an adaptive management strategy with step by step monitoring and eventual corrections if needed.

## 7   Conclusion

The evolution of river morphology is very complicated to predict, especially in the case of mountain and Piedmont rivers with complex morphologies. River restoration in such terrain can thus be challenging for river managers due to the random nature of riverbed evolution. Reliable hydromorphological numerical modelling combined with good field expertise can be helpful in this case for better river management. Within this framework, our study focused on the development of a 2D hydromorphodynamic model over the "Lac des Gaves" reach in the Hautes-Pyrénées, France, with the TELEMAC-MASCARET system. This river

reach has precisely the morphological characteristics mentioned above as it is a braided channel with a very heterogeneous grain-size distribution. The aim was to reproduce the channel evolution following the 2018 flood event that considerably impacted the channel's morphology to propose relevant and sustainable restoration solutions. Two bedload transport formulas, the Meyer-Peter-Müller and the Recking one, were used with the Ferguson and the Strickler friction laws to assess sediment transport processes. Three performance criteria were considered to assess the validity of the developed model: the Brier Skill

Score, the comparison of longitudinal profiles and the analysis of deposited volumes within the LDG.

    The 2D hydromorphodynamic model performed realistic simulations with the Ferguson friction law for both sediment transport formulas (Recking and MPM). These results validate the necessity to use a friction formula adapted to river reaches with high relative roughness and significant sediment load. The developed model tends to overestimate sediment depositions within the LDG. This might be due to the fact that it is a monodisperse model, considering bedload only with one homogeneous

grain-size whereas in reality, finer sediments are also available. These are likely to be "flushed" and travel longer distances before being deposited, which is not simulated here. The observed modelled morphological changes can thus be considered to overestimate what can actually be noticed on the ground. Further improvements on these aspects are necessary, knowing the heterogeneity of sediment sizes within the LDG reach. Simulations on the updated sediment transport module GAIA, developed to handle the grain-size distribution issue better, are considered to improve the hydromorphodynamic model.

Moreover, this study shows that the BSS might not be the right performance criterion to consider for rivers with braided morphologies. These complex configurations remain very difficult to reproduce by 2D models. The BSS score can thus give very pessimistic results, whereas the model correctly reproduces the most important processes (erosion and deposition areas). We recommend considering an integrative approach where the modeller combines multiple assessment criteria such as long profiles and cross-sections evolution and volumetric estimations to judge the model's performance.

Finally, even if our model can still be improved, it provided valuable information on the possible consequences of a restoration scenario to river managers. Many operational issues were raised for the weir lowering scenario, such as the increase of flood risks downstream or severe erosions upstream that can translate the issue to the upstream weir. Knowing the complexity of river restoration projects in these kinds of complex morphologies, considering an adaptive management strategy with a step by step monitoring and eventual corrections might be more appropriate rather than a radical measure. Besides, enhancing

the hydromorphodynamic model after considering the whole grain-size distribution and its actualization after each morphogenetic event can be used as a decision-making tool that can assist river managers and help them communicate with the elected representatives.

*Author contributions.* Rabab Yassine carried out the simulations and the field experiments in collaboration with Hélène Roux, Ludovic Cassan and Olivier Frysou. Rabab Yassine wrote the manuscript with support from Ludovic Cassan and Hélène Roux. François Pérès helped
supervise the project and organize stakeholders meeting to communicate the results with the elected representatives.

*Competing interests.* No competing interests are present.

*Acknowledgements.* We are grateful to the public institution "Pays de Lourdes et des Vallées des Gaves" for having given us the opportunity to undertake this work. This research was funded by the three following organisms: the Water Agency "Adour-Garonne", the government, the region "Occitanie Pyrénées Méditerranée" whom we thank sincerely.

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
