# Peer review of "Numerical modelling of the evolution of a river reach with a complex morphology to help define future sustainable restoration decisions"

_Earth Surface Dynamics, 2021_

## Referee Comment (RC2)

Review of "Numerical modelling of the evolution of a river reach with a complex morphology to help define future sustainable restoration decisions" by Yassine et al. (2022)

Reviewer: Clément Misset, Grenoble, France - March 30, 2022.

General comments

This manuscript presents the 2D numerical modelling of morphodynamics in a braided river reach during a large flood event. To develop and test the 2D model, it uses observed morphological changes (DoD obtained from Lidar data) as well as hydrological data and field measurements. This is thus a nice dataset. The manuscript is generally well written and addresses important questions for both researchers and practitioners, regarding the modeling of morphodynamics in braided systems. Several friction laws and bedload transport formulae are tested in the numerical model in order to evaluate the most relevant modeling approach. Criteria to evaluate the model performance in such context are discussed as well as future improvements. I think this work is of interest for the reader of Earth Surface Dynamics.

I have however several concerns, that I suggest to consider in a revised version of the manuscript:

(1)  I think some relevant literature is missing, in particular regarding braided morphodynamics modeling and criteria that have already been proposed to evaluate 2D morphodynamics models (see detail comments).

(2)  More details could be added when presenting the studied site as well as some of the model parameters, for the reader to better understand the setting and modeling choices (see detail comments).

(3)  Concerning the sediment transport deposition partitioning, I have a major concern: can the historical partitioning observed with a lake configuration be used for the braiding setting? I guess this is not the case. Using the same values (8-16% of bedload) should be discussed regarding the hydrodynamics observed/modeled, typical values observed in previous works, etc. This assumption seems to be critical, in particular for the model evaluation.

(4)  The choice (critical) of the upstream boundary condition for bedload flux would beneficiate to be better argued.

(5)  Concerning the performance criteria, I think the paper would greatly beneficiate to include other criteria adapted to braided rivers. The statistical distribution of erosion and disposition have for instance been used in a similar study by Williams et al. (2016a). There are also several braiding index that have been used by Rifai et al. (2014). These aspects are already discussed in the current manuscript, but I think using more appropriate index could significantly improve the analysis.

(6)  It could be considered to add a discussion section, to better distinguish the results description and their interpretation.

To sum up, I think this work is original and will be of great interest for geomorphologists. I however think that substantial revision could be made to improve the manuscript regarding the previously mentioned aspects.

Detail Comments

Introduction: I suggest to better define "complex" morphology. I also suggest stating sooner and more clearly the main objectives of the paper, for the reader to understand why a numerical model is developed.

L19: Sentence is not clear, are you citing Rickenmann et al. (2016) or Reisenbüchler et al., 2019? Consider rewrite the sentence

L27: the term "condition" is vague, be more specific. Natural hazards have been already discussed in previous lines; I suggest removing it from this sentence

L36: I suggest adding that hydrodynamics is also driven by morphological changes

L41: using valid sediment transport formulae (friction law, etc.) is not only needed to introduce reliable boundary conditions but to develop realistic modeling "inside" the model domain (at the grid scale).

L65: I think there is some relevant literature about braided morphodynamics modeling that is missing. I suggest considering Williams et al. (2013), Williams et al. (2016a), Williams et al. (2016b), Rifai et al. (2014), Gonzales de Linares et al. (2021). Comparison with these works would also be relevant in the discussion section.

Study area description: I think it would be of interest for the reader to add a more detailed description of the site, in particular, including a river profile showing slopes, the tributary, solid material input, etc. If available, I suggest to use a hillshade map of the catchment in Fig.1 (instead of the grey areas) to better show the relief and the mountainous environment. Maybe also add pictures illustrating the upstream main tributary morphology?

Fig2: I suggest writing directly on the picture the dates and mention "before the 2013 flood", "after the 2013 flood".

L104: "compared to 90 m3/s annually in the same period" -> do you mean: compared to the monthly averaged discharge of 90m3/s?

2.4 Restoration implications: I have a concern regarding the model aim. I am not convinced that once calibrated on a rare event such a the 2013 event, the model will automatically be applicable for long term modeling, in particular for low magnitude events that would likely contribute to transport the stored sediments from the LDG to the downstream reaches in case the weirs are removed.

L128: I would add something like "Thus, to help decision makers, a hydro-morphological 2D model…"

3.2.1 Friction laws: specify that the friction law is not coupled with the bed surface grain size (which is constant because fractional transport is not considered in the morphodynamical modeling).

L170: I wouldn't say "The friction coefficient for the Ferguson (2007) law is the D84", I would say something like "the Ferguson law uses the D84 as a proxy of the bed roughness" or remove this sentence

L185: I found the sentence not clear, K/K' is used for shear stress partitioning if I'm right. Do you use this shear stress partitioning in your 2D modeling? According to Gonzales de Linares et al. (2020, 2021), such correction might not be relevant in massive bedload transport.

The recking formula: I have a major concern here. You use a 1D "morphological" formula that takes implicitly into account the cross-section averaging (variability of shear stress, grain size distribution, etc.) to deal with non-linearity effect of bedload transport. In your 2D model, the shear stress is spatially discretized so that the Shield number used to calculate the bedload flux is a "local" Shield number and the bedload transport formula does not need to take into account cross section averaging. I suggest considering the following papers: Recking (2013) and Recking et al. (2016).

3.2 Sediment transport and bed evolution module: Could you specify if an avalanche mode (Slope sliding) is used and what are the parameter considered (angle of repose)? Same question regarding the deviation of sediment transport on transverse slopes? Both could have significant effect on the morphological modelling, it is thus of interest for the reader to know which parameters were used.

L223-224: could you provide the point density of the Lidar and the raster resolution of the used DEMs? Could you specify here if bathymetric data for the low flow channels were available?

L235: It is not clear for me how you estimate the bedload fraction? Do you have GSD measurement of the dredging? This should be clarified. The range 8-16% seams possible but much wider range could be considered, see for instance Turowski et al. (2010). I also wonder if you can consider the same partitioning between the pre and post flood situation: Fines will likely deposit more easily in a lake (even if they will deposit in the braiding situation). Note that the sediment transport partitioning might also evolved with the event return period and the flow/sediment supply conditions so that dredging might not be fully representative of a specific event. The following papers could be of interest to consider a gravel matrix fine fraction commonly observed in gravel bedded stream: Mueller and Pitlick (2013), Navratil et al. (2010), Misset et al, (2020)

4.1.2 Input hydrograph: do you have points with liquid discharge measurement to calibrate/validate the model? This should be specified.

Fig5. It is not clear for me if the large model uses a fixed bed? I suggest to add a title above each mesh (large, finer mesh).

L277: the equilibrium load upstream condition: Is this relevant? Were the upstream profile and cross section stable? Did you compare these upstream fluxes with an averaged 1D formula? This hypothesis is probably critical for the modeling of the event.

L285: I suggest specifying that under the hydrodynamics calibration conditions, morphological changes and bedload transport were limited. If bathymetric data were not available, this can probably lead to a non-negligeable uncertainty on the water surface elevation, which can be considered acceptable compared to other uncertainties, this need to be specified.

Also, did you vary the D84 considered in the Ferguson friction law? Or did you use the measured value?

L305-306: maybe use the term DoD (DEM of differences)?

4.4 Performance evaluation: Braiding and morphological changes in braided rivers are somewhat stochastic processes. Is it thus relevant to use single long profile/cross sections and the Brier Skill Score objective function? I mean, your model can be considered relevant in reproducing the morphodynamics of the studied reach as it reproduces well some general morphological evolution (for instance the braiding index, the number of active channels, the statistical distribution of erosion and deposition, the average long profile, etc.) but the objective function can lead to a bad score (you do not have exactly the same DEM evolution).

4.4.3 Comparison of the deposited volumes

Comparison of the global volumetric budget of the reach is of great interest. I have however the filling that DoD obtained for the 2018 flood includes much more information. You could for instance compare that statistical distribution of erosion/deposition in the reach, is is done in a similar study by Williams et al. (2016a).

Fig7: I suggest using red for erosion and blue for deposition (commonly used colors for such map), to include a scale bar and to remove the pixel within the uncertainty range of the DoD (by using transparent color so that only significant morphological differences can be seen).

L349: Friction laws link velocity to depth and roughness

5.1 General visual comparison of eroded and deposited areas: according to a previous comment, you are comparing visually the erosion and deposition distribution depending on the friction law and transport formula used: comparing these distributions could strengthen this analysis.

Another aspect that could be discussed regarding the effect of the friction law is the use of spatially and temporally constant roughness parameters (n or D84). It is likely that these parameters are not constant in both space and time the braiding reach, which can explain some of the differences between modeled and measured evolutions.

L372: If I understand, the model seems to be not fully relevant in the downstream part of the reach, as almost no evolution is modeled. I agree that such model, already complex, doesn't take into account all the complexity of such braided system. I however wonder if this disagreement is only due to the fact that suspended load is not modeled. Do you have observations showing that the downstream part is mainly driven by suspended load processes? Based on aerial photograph, the downstream part of the reach seems to be at least partially composed of gravel bars? I suggest moving this discussion of the results in the discussion section.

Fig.8 and 9: I suggest adding on the map "observed changes" and the names of the transport formula and friction law.

5.2 Longitudinal profiles and cross-section comparison: I wonder if comparing long profile in braided morphology is relevant? Maybe, using an "averaged" long profile to capture the general trend and being less sensitive the the active channel position would be a better option? I also suggest to move from this section all sentences discussing this specific point and the relevance of the BSS index.

L415: Can we consider the deposited bedload fraction is similar between the 2018 braiding situation and the pre 2013 flood situation with a lake? I have the feeling that the deposited suspended load fraction would be higher in a lake configuration. Were the hydrodynamic conditions low enough (backwater effect, low velocities, low shear stress, etc.) during the 2018 flood so that more than 80% of the deposited volume corresponds to suspended particles? If it is not the case, this fraction seems to be really high. I suggest comparing it with previous works on fine particle stocks/content in gravel bedded streams (see Mueller and Pitlick (2013), Navratil et al. (2010), Misset et al, (2020)).

References:

Williams,R.D., J. Brasington, M. Hicks, R. Measures,C.D.Rennie, andD.Vericat (2013),Hydraulic validation of two-dimensional simulations of braided river flow with spatially continuous aDcp data, Water Resour. Res., 49, 5183–5205, doi:10.1002/wrcr.20391.

Williams, R. D., R. Measures, D. M. Hicks, and J. Brasington (2016a), Assessmentof a numerical model to reproduce event-scale erosion and deposition distributions in a braided river, Water Resour. Res., 52, 6621–6642, doi:10.1002/2015WR018491.

Williams, Brasington and Hicks, Numerical Modelling of Braided River Morphodynamics: Review and Future Challenges (2016b), Geography Compass 10/3 (2016): 102–127, 10.1111/gec3.12260 Numerical

Ismail Rifai, Caroline Le Bouteiller Alain Recking (2014), Numerical Study of Braiding Channels Formation, 21st Telemac & Mascaret User Club, Grenoble, France, 15-17 October, 2014.

Gonzales de Linares, Florian Ronzani, Alain Recking, Vincent Mano, Guillaume Piton (2021), Coupling surface grain-size and friction for realistic 2D modelling. SimHydro 2021: Models for complex and global water issues - Practices and expectations 16-18 June 2021, Sophia Antipolis.

Gonzales de Linares, Alain Recking, Vincent Mano, Guillaume Piton (2020), Modelling of massive bedload deposition in a debris basin: cross comparison between numerical and small-scale modelling in River Flow 2020: Proceedings of the 10th Conference on Fluvial Hydraulics (Delft, Netherlands, 7-10 July 2020), Uijttewaal et al (eds) ISBN 978-0-367-62773-7. pp. 282–289. 2020.

Recking (2013), An analysis of nonlinearity effects on bed load transport prediction, JOURNAL OF GEOPHYSICAL RESEARCH: EARTH SURFACE, VOL. 118, 1–18, doi:10.1002/jgrf.20090, 2013

Alain Recking, Guillaume Piton, Daniel Vazquez-Tarrio and Gary Parker (2016), Quantifying the Morphological Print of Bedload, Earth Surf. Process. Landforms (2016) Copyright © 2016 John Wiley & Sons, Ltd. Published online in Wiley Online Library (wileyonlinelibrary.com) DOI: 10.1002/esp.3869

JENS M. TUROWSKI, DIETER RICKENMANN and SIMON J. DADSON (2010), The partitioning of the total sediment load of a river into suspended load and bedload: a review of empirical data, Sedimentology (2010) 57, 1126–1146, doi: 10.1111/j.1365-3091.2009.01140.x

Mueller, E. R., & Pitlick, J. (2013). Sediment supply and channel morphology in mountain river systems: 1. Relative importance of lithology, topography, and climate, Journal of Geophysical Research: Earth Surface, 118(4), 2325–2342. https://doi.org/10.1002/2013jf002843

Navratil, O., Legout, C., Gateuille, D., Esteves, M., & Liebault, F. (2010). Assessment of intermediate fine sediment storage in a braided river reach (southern French Prealps). Hydrological Processes, 24(10), 1318–1332. https://doi.org/10.1002/hyp.7594

C.Misset, A.Recking, C.Legout, B.Viana-Bandeira, A.Poirel (2020), Assessment of fine sediment river bed stocks in seven Alpine catchments, https://doi.org/10.1016/j.catena.2020.104916

---

## Referee Comment (RC3)

[referee-annotated manuscript omitted]

---

## Author Comment (AC1)

**Answer to RC1: Comment on esurf-2021-91 from Damien Kuss**

June 26, 2022

We wish to thank the referee for his careful evaluation of the manuscript. Please find below the detailed answers. The reviewer's comments are shown in bold and some modifications of the manuscript are emphasized in blue.

**General comments**

**This article is original and particularly interesting with the use of a 2D hydro-morphological numerical model in a torrential context. The task is challenging as the study concerns the modeling of a reach of braided river with wandering flows which are by nature random and are therefore particularly difficult to model in a deterministic way.**

**From a methodological point of view, the numerical aspects are very well described. On the other hand, the study site and the modeled domain deserve to be better described: length, slopes... A longitudinal profile, encompassing the modeling domain, would make it possible to better understand the problem of deposition during floods in link with the weirs and with the decrease in the longitudinal slope. The influence of the solid volume taken inot account, linked with the slope, could be discussed in the results section.**

A better description of the study site including length and slopes will be added in the text. The longitudinal profiles extracted from the LIDAR DEM of 2016 and 2019 will be plotted in the description part (Fig. 1).
A discussion will be added in the boundary conditions section regarding the possible influence of the upstream boundary condition on solid discharge, as explained below.

**The hydromorphological modeling is carried out taking into account only bed-load (with Meyer Peter and Recking formulas). But you mention, by exploiting the data from dredging, that the fraction of the volume transported by bedload would represent only 8 to 16% of total transport (lines 417 to 417). There is an inconsistency:**

- **The altitudinal evolutions observed by DoD are compared to the modeled bed evolutions, which only include sediment transport by bedload;**

- **On the other hand, the modeled transported volume is compared to the total volume observed minus the fine fraction (volume transported by suspension).**

In the Lac des Gaves, the deposit consists of an upper layer of coarse material over the first few centimeters, with finer sediment stored below. In the simulations, only the bedload has been taken into account while the difference in altitude between the LIDAR DEM of 2016 and 2019 takes into account the total load. This is one of the reasons why several differences are observed between simulations and DEM evolution. Considering only bedload transport is probably the most wrong in the downstream area of Lac des Gaves where significant deposition of fine sediments has been observed. These aspects have been mentioned in the manuscript already but will be clarified and detailed.
As far as volumes are concerned, the simulations were compared to the dredging volume, but

[Figure]

Figure 1: Longitudinal profile evolution following the flood of 2018.

taking into account only a fraction of this dredging volume: between 8 and 16%, which refers to the fraction of bedload transport. This will also be clarified in the text.

**The performance of each modeling scenario is evaluated with the BSS score. In a braided river context, the scores obtained are not good (BSS = 0.06). You discuss the representativeness of this metric for such river morphologies. If the metric can be criticized because of the random nature of the wanderings, it would also seem relevant to question the added value of a 2D model compared to a 1D model in such a context. You could precise the configurations where a 2D model is apropriate.**
Wouldn't a 1D model have been enough? The choice of a 2D model has been made because it allows a better representation of the hydrodynamics and in particular of the friction with taking into account a spatialization of the water height. Even if the representation of the braiding and of the different flow arms is not the real one, the 2D model has the advantage of a continuity of the dynamics, contrary to the 1D model with interpolation between two profiles and water height projected on the DEM to estimate the extent of the flooded area.
For all these reasons and despite the random nature of the wandering which is difficult to reproduce, we still think that 2D is a better choice than 1D in the Lac des Gaves area. On the other hand, the choice of the BSS criterion is more debatable. Such a discussion will be added in the model section regarding the 1D vs 2D choice and in the final discussion section regarding the BSS choice.

**Specific comments Reference Reisenbücchler et al., 2019 at the end of sentence line 19. What does it refers to ?**
This is a typo, this reference will be removed.

**Line 19 : "They showed". "They" refers to Rickenmann et al. or Reisenbuchler et al. ?**
"They showed" will be replaced with "Reisenbüchler et al. (2020) showed".

**Line 38 to 40 :**

- **a mention could be made of more recent formulas partially based on field data (Recking, 2013 ; Lefort, 2015).**
  The Recking formula will be mentioned at that point. We didn't find a formula from Lefort in 2015 but we will mentioned the one from Lefort (2007).
  although more recent formulas are partially based on field data (Recking, 2013; Lefort, 2007)

- **are bedload transport formulas established in 1d narrow channels directly transposable in 2D models ?**
  The Recking formula that has been implemented is a version compatible with 2D calculation. It will be added in the text.
  We used the version of this formula compatible with 2D calculation and local data (Recking et al., 2016) which is introduced below.

$$\phi = 14 * \frac{\tau^{*\ 2.5}}{1 + (\frac{\tau_m^*}{\tau^*})^{10}} \tag{1}$$

with

$$\tau_m^* = 0.26 * S^{0.3} \tag{2}$$

The parameter $\tau_m^*$ is a mobility term that defines the transition between partial transport and full mobility Recking et al. (2016), $\tau^*$ is the Shields number and $S$ [m/m] the river bed slope.

**Lines 40-41 : the slope used for the sediment transport calculation at the upstream boundary condition of the model is a key parameter to perform realistic simulations. It must be apprehended by a geomorphological analysis based on the the longitudinal profile.**
As explained, a morphological equilibrium condition has been set at the inlet because of numerical instabilities generated at the level of the boundary cells with other kinds of solid boundary conditions. We agree that this is indeed a strong assumption and studies are currently underway with the new solid transport module Gaia of the TELEMAC-MASCARET modelling system to test the impact of this boundary condition on the simulations.

**Line 63 : "The TELEMAC-MASCARET modelling system has been considered well suited to perform 2D morphodynamic simulations on the LDG reach". Why ?**
Clarifications will been added to the text.
Indeed, previous studies have shown that TELEMAC/Sisyphe was able to reproduce processes of erosion/deposition accurately in similar configurations (Reisenbüchler et al., 2020, 2019; Cordier et al., 2019). Sisyphe enables the use of different transport formulas (Meyer-Peter and Müller, 1948; van Rijn, 1984) and also take into account various factors influencing sediment transport, such as the effect of the bed slope (Koch and Flokstra, 1981; Soulsby, 1997) on the magnitude of the bedload transport (Riesterer et al., 2016). It also offers the possibility of programming other formulas, both for the parameterisation of friction and for solid transport, a possibility which has been used here to introduce formulations more adapted to the context of mountain rivers.

**Line 92-93 : you could add a reference at the end of the sentence.**
Reference will be added in the text.

**a longitudinal profile would complete the description and would allow to better understand the effects of the two weirs during the floods.**

The longitudinal profiles extracted from the LIDAR DEM of 2016 and 2019 will be plotted in the description part.

**Lines 101-104 : you could add references concerning the peak discharges.**
Reference will be added in the text.

**Figure 3 : you could add the reference for each picture**
References will be added in the text.

**Lines 164-166 : you could add a reference : Rickenmann and Recking (2011) ?**
Reference will be added in the text.

**Lines 185-188 : please define all the terms used in equation 5**
The definitions of $\Phi$ and $\tau*$ will be added in the manuscript.
$\Phi$ is the dimensionless solid transport, calculated as $\Phi = \frac{q_{sv}}{\sqrt{g(\rho_s/\rho-1)D^3}}$ with $q_{sv}$ $[m^3/s/m]$ the unit solid volume transport: $q_{sv} = Q_{sv}/W$ with $Q_{sv}$ $[m^3/s]$ the solid volume flow rate, $W$ $[m]$ the river width, $\rho_s$ $[kg/m^3]$ the density of the sediments, $\rho$ $[kg/m^3]$ the density of water and $D$ $[m]$ the grain diameter.

$\tau*$ $[-]$ is the Shield number, calculated as $\tau* = \frac{\tau}{g(\rho_s-\rho)D}$ with $\tau$ $[N/m^2]$ the shear stress.

**Lines 195-208 : note that the recking formula : (a) is mainly base on lab experiments for high shields numbers ; (b) is very sensitive to the choice of the shields mobility parameter. So the avantages of using this formula for intense floods where the shields number exceeds the mobility parameter could be discussed.**
Clarifications will be added in the text.
The parameter $\tau_m^*$ is a mobility term that defines the transition between partial transport ($\tau^* < \tau_m^*$) and full mobility ($\tau^* > \tau_m^*$) (Recking et al., 2016). The Recking formula was calibrated on field data ($\tau^* < \tau_m^*$) and laboratory data ($\tau^* > \tau_m^*$). It is the value of $\tau_m^*$ that gives its shape to the model. Therefore the value of $\tau_m^*$ strongly impacts the result, and its determination is difficult, especially for mountain streams. Ideally it should be based on measurements. Failing that, the available data suggest that an estimate is possible (Recking et al., 2016).

**Lines 225-226 : it is not clear how you use the dredging data.**
Clarifications will be added in the text.
Indeed, coarse sediment dredging data over 11 years were collected upstream the first weir. Using this data, the fraction of bedload of the total sediment transport has been estimated. This is necessary as our model only consider bedload transport when in fact there is both bedload transport and transport in suspension.

**Lines 234-235 : "However, the recorded volumes represent both very fine sediments probably transported by suspension and very coarse sediment via bedload transport". The model used in the study takes into account only the bedload transport, Is it right ?**
Yes, that's right, the numerical model only takes into account the bedload transport but only 8-16% of the dredging volumes are compared to the simulated volumes as explained in section 5.3. The 8-16% range corresponds to the estimated contribution of bedload to total transport.

**Lines 269-270 : you therefore make the assumption of no downstream influence.**
Yes that's right, we made the assumption of no downstream influence, that is to say dewatered regime for the weir which is most often verified on site.

**Line 276 : what do you mean by "instabilities" ?**
These are numerical instabilities that lead to aberrant erosion or deposit, extremely high and localized on only one or two cells of the boundary. Clarifications will be added to the text.
*Unfortunately, this generated many numerical instabilities that lead to aberrant erosion or deposit, extremely high and localized on only one or two cells of the upstream boundary.*

**Lines 276-278 : the slope of the LDG reach is never given. What is slope at the upstream boundary condition ? Is the hypothesis of stable riverbed evolution justified ? What was the observed evolution of the river bed at the upstream condition during the floods ?**
The bed slope at the upstream boundary condition is 0.018 $m/m$. As mentionned before, it was difficult from a numerical point of view to impose solid discharge from larger model. Nevertheless, the area of interest of the study is located between the 2 weirs (between 500 and 2000 m on figure 10). In this part, the evolution of the bed is less influenced by the choice of the boundary condition than in the upstream part. Then it can be assumed that the upstream boundary condition on solid discharge has low influence in the area of interest: the upstream condition is located sufficiently far from it to reduce its influence. This is a relatively good assumption for the flood event of 2018 for which little material seems to have come from upstream the area of interest. Of course this will not be true for the flood event of 2013 for which large amounts of sediments have come from upstream. Again, we agree that this is indeed a strong assumption and studies are currently underway with the new solid transport module of the TELEMAC-MASCARET modelling system to test the impact of this boundary condition on the simulations. Discussions will be added in the text on this topic.
*At that location, the bed slope is 0.018 $m/m$ (Fig. 6). The particularity of this boundary condition is that it delivers sufficient bedload at the model inlet to keep the riverbed elevation at the inlet cross-section constant in time. It has been assumed that the upstream boundary condition on solid discharge has low influence in the area of interest which is the Lac des Gaves: the upstream condition is located sufficiently far from it to reduce its influence. This is a relatively good assumption for the flood event of 2018 for which little material seems to have come from upstream the area of interest.*

**Lines 286-287 : you could detail what you mean by numerical and physical parameters.**
The numerical parameters are related to the time step, the type of solver and its accuracy for instance. The sentence will be removed as it's not the purpose of the paper and it may be confusing.

**Figure 6 : how do you transform 2D results into 1D longitdunal profile ?**
There is no transformation: the 1D plot is just an extraction of the longitudinal profile from the lowest bathymetric points of the 2D results. Clarification will be added in the text.
*The 1D longitudinal profiles of the present paper are drawn from an extraction of the lowest bathymetric points of the 2D model.*

**Lines 326-327 : "To date, numerical models cannot predict channel migration processes that occur in braided rivers. These phenomena are uncertain and random. A modeler should thus not expect the model to predict channel migration accurately during a flood.". I agree. Butin consequence you should better justify why you have chosen a 2D hydromorphological model for this case study.**
Clarification on the choice of a 2D hydromorphological model will be added in the text, in the model description section.

The choice of a 2D model has been made because it allows a better representation of the hydrodynamics and in particular of the friction with taking into account a spatialization of the water height. Even if the representation of the braiding and of the different flow arms is not the real one, the 2D model has the advantage of a continuity of the dynamics, contrary to the 1D model with interpolation between two profiles and water height projected on the DEM to estimate the extent of the flooded area.

**Lines 331-332 : is the 2019 LiDAR realigned ?**

No the 2019 DEM is not realigned, Fig. 7a was obtained by calculating the difference in elevation between the 2 available DEMs, the one of 2016 and the one of 2019.

**Figures 10-11 : I regret that the 2016 profile was not drawn with a solid line. it's hard to see the position of the fall. You could also explain how you transform 2D results with a bed level not constant over cross sections in to longitudinal profiles.**

The 2016 profile will be drawn with solid line. There is no transformation of the 2D results: the 1D plot is just an extraction of the longitudinal profile from the lowest bathymetric points of the 2D results.

**Figure 13 : how do you explain the deposits above the max water level ?**

There is no deposition above the maximum water level. The elevation at 439 m at about 130 m cumulated distance corresponds to the topography from the original 2016 DEM that has not been eroded.

Clarifications will be added to the text.

For instance, the elevation at around 439 $m$ at about 130 $m$ cumulated distance in the upstream cross-section (Fig. 13a) and 13b)) corresponds to the topography from the original 2016 DEM that has not been eroded during the simulation.

**Figure 14 : the initial profile is missing. It could be a usell information.**

The initial cross sections are already plotted, it is the 2019 DEM which is the most recent topography information available from which the restoration scenarios have been simulated (Z_2019 on the figure). Clarifications have been added to the text.

Two restoration scenarios were performed using the LiDAR DEM surveyed in 2019 as the initial topography.

and in the legend of Figure 14.

Z_2019 represents the LiDAR DEM surveyed in 2019 that is the initial topography.

**References**

Cordier, F., Tassi, P., Claude, N., Crosato, A., Rodrigues, S., and Pham Van Bang, D.: Numerical Study of Alternate Bars in Alluvial Channels With Nonuniform Sediment, Water Resources Research, 55, 2976–3003, https://doi.org/https://doi.org/10.1029/2017WR022420, 2019.

Koch, F. and Flokstra, C.: Bed level computations for curved alluvial channels, in: XIXth Congress of the International Association for Hydraulics Research; New Delhi, India, 1981.

Lefort, P.: Une formule semi-empirique pour le transport solide des rivières et des torrents, in: Colloque SHF - Transport solide et gestion des sédiments en milieux naturels et urbains, 2007.

Meyer-Peter, E. and Müller, R.: Formulas for Bed-Load transport, IAHSR 2nd meeting, Stockholm, appendix 2, publisher: IAHR, 1948.

Recking, A.: Simple method for calculating reach-averaged bed-load transport, Journal of Hydraulic Engineering, 139, 70–75, 2013.

Recking, A., Piton, G., Vazquez-Tarrio, D., and Parker, G.: Quantifying the Morphological Print of Bedload Transport, Earth Surface Processes and Landforms, 41, 809–822, https://doi.org/10.1002/esp.3869, 2016.

Reisenbüchler, M., Bui, M. D., Skublics, D., and Rutschmann, P.: An integrated approach for investigating the correlation between floods and river morphology: A case study of the Saalach River, Germany, Science of The Total Environment, 647, 814–826, https://doi.org/10.1016/j.scitotenv.2018.08.018, 2019.

Reisenbüchler, M., Bui, M. D., Skublics, D., and Rutschmann, P.: Enhancement of a numerical model system for reliably predicting morphological development in the Saalach River, International Journal of River Basin Management, 18, 335–347, https://doi.org/10.1080/15715124.2019.1628034, publisher: Taylor & Francis, 2020.

Riesterer, J., Wenka, T., Brudy-Zippelius, T., and Nestmann, F.: Bed load transport modeling of a secondary flow influenced curved channel with 2D and 3D numerical models, Journal of Applied Water Engineering and Research, 4, 54–66, https://doi.org/10.1080/23249676.2016.1163649, publisher: Taylor & Francis _eprint: https://doi.org/10.1080/23249676.2016.1163649, 2016.

Soulsby, R.: Dynamics of Marine Sands: A Manual for Practical Applications, Thomas Telford, London, 1997.

van Rijn, L. C.: Sediment Transport, Part II: Suspended Load Transport, Journal of Hydraulic Engineering, 110, 1613–1641, https://doi.org/10.1061/(ASCE)0733-9429(1984)110:11(1613), 1984.

---

## Author Comment (AC2)

**Answer to RC2: Comment on esurf-2021-91 from Clément Misset**

June 28, 2022

We wish to thank the referee for his careful evaluation of the manuscript. Please find below the detailed answers. The reviewer's comments are shown in bold and some modifications of the manuscript are emphasized in blue.

**General comments**
**This manuscript presents the 2D numerical modelling of morphodynamics in a braided river reach during a large flood event. To develop and test the 2D model, it uses observed morphological changes (DoD obtained from Lidar data) as well as hydrological data and field measurements. This is thus a nice dataset. The manuscript is generally well written and addresses important questions for both researchers and practitioners, regarding the modeling of morphodynamics in braided systems. Several friction laws and bedload transport formulae are tested in the numerical model in order to evaluate the most relevant modeling approach. Criteria to evaluate the model performance in such context are discussed as well as future improvements. I think this work is of interest for the reader of Earth Surface Dynamics.**
**I have however several concerns, that I suggest to consider in a revised version of the manuscript:**

1. **I think some relevant literature is missing, in particular regarding braided morphodynamics modeling and criteria that have already been proposed to evaluate 2D morphodynamics models (see detail comments).**

2. **More details could be added when presenting the studied site as well as some of the model parameters, for the reader to better understand the setting and modeling choices (see detail comments).**

3. **Concerning the sediment transport deposition partitioning, I have a major concern: can the historical partitioning observed with a lake configuration be used for the braiding setting? I guess this is not the case. Using the same values (8-16% of bedload) should be discussed regarding the hydrodynamics observed/modeled, typical values observed in previous works, etc. This assumption seems to be critical, in particular for the model evaluation.**

4. **The choice (critical) of the upstream boundary condition for bedload flux would beneficiate to be better argued.**

5. **Concerning the performance criteria, I think the paper would greatly beneficiate to include other criteria adapted to braided rivers. The statistical distribution of erosion and disposition have for instance been used in a similar study by Williams et al. (2016b). There are also several braiding index that have been used by Rifai et al. (2014). These aspects are already discussed in the current manuscript, but I think using more appropriate index could significantly improve the analysis.**

**6. It could be considered to add a discussion section, to better distinguish the results description and their interpretation.**

**To sum up, I think this work is original and will be of great interest for geomorphologists. I however think that substantial revision could be made to improve the manuscript regarding the previously mentioned aspects.**

1. We will add relevant literature as proposed by the reviewer. The details are given below in several detailed comments (e.g. L65, L415) and in the references.

2. A better description of the study site including length and slopes will be added in the text. A plot of the longitudinal profiles extracted from the LIDAR DEM of 2016 and 2019 will be added in the description part. The figure that we intend to add to the last version of the article is presented below (Fig. 1).

[Figure]

Figure 1: Longitudinal profile evolution following the flood of 2018.

3. This question is very relevant and we understand the concern about using historical dredging data to estimate the part of bedload transport within the study area. However, we consider that the site configuration is intermediate between a lake and a braided river since the slope is lower than the equilibrium one because of the downstream weir. This decelerated flow is observed in the major part of the domain where sediment transport is analysed in detail (between the two weirs). As a consequence, if we take the value for the lake, we observe that the part of the bedload is close to the upper interval limit because the hydrodynamic forces are actually larger than within the lake. The conclusions about bedload transports as a function of the models are not influenced by this assumptions. Concerning comparison with experiments, the analysis is completed by profile in the following.

4. The upstream condition is indeed a crucial point. As mentioned, imposing solid discharge from a reach-averaged equation was difficult and generated numerical instabilities at the level of the boundary cells with other kinds of solid boundary conditions. Nevertheless, the area of interest is between the two weirs (between 500 and 2000 m in figure 10). In this part, the bed evolution is less influenced by models than the upstream part. Then it can be assumed that the solid upstream boundary condition has low influence in the area of interest. The upstream part of the domain is used to create a more realistic solid discharge.

Clarifications will be added to the text.
The upstream boundary condition is located sufficiently far from the area of interest to influence its dynamics directly.

5. As suggested, we will include the statistical distribution of erosion and deposition used by Williams et al. (2016b) (See Fig. 2 upstream the LDG and Fig. 3 within the LDG). Compared to the measurements, it can be seen that some models reproduce the spatial distribution of deposited or eroded areas better than others, this will be further discussed in the article to improve the analysis.

6. A discussion section will be added in the article.

**Detail Comments**

**Introduction: I suggest to better define "complex" morphology. I also suggest stating sooner and more clearly the main objectives of the paper, for the reader to understand why a numerical model is developed.**

By complex morphology, we mean a river with many channels and very heterogeneous grain-size distribution. It is not a single channel with uniform materials. To better present the interest of numerical model, we will add clarifications at the beginning of the introduction:

Numerical models allows taking into account complex geometry with several braid river but also various class of sediment. Generally they are necessary in case of several physical models have to be considered. For instance, they provide velocity, suspended concentration of grain size transported which has to be known for ecological purpose. For flood impact forecast, they are able to estimate time scale of erosion or deposition. They can also evaluate morphogical evolution in area where the expertise is lacking as close to hydraulic structure with specific design.

**L19: Sentence is not clear, are you citing Rickenmann et al. (2016) or Reisenbüchler et al., 2019? Consider rewrite the sentence**

"They showed" will be replaced with Reisenbüchler et al. (2020) showed.

**L27: the term "condition" is vague, be more specific. Natural hazards have been already discussed in previous lines; I suggest removing it from this sentence**

We will precise the term and remove the redundant following sentence.

Besides safety issues, bedload transport, combined with water discharge, is considered a fundamental driver of river morphodynamics and risks of overflowing.

**L36: I suggest adding that hydrodynamics is also driven by morphological changes**

We will add a sentence that underlines the back influence of morphology on hydrodynamics.

Simultaneously, the morphological modifications have then an influence on the hydrodynamic simulation.

**L41: using valid sediment transport formulae (friction law, etc.) is not only needed to introduce reliable boundary conditions but to develop realistic modeling "inside" the model domain (at the grid scale).**

We will rewrite the sentence which was confusing about the need to have robust modelling of the physical phenomena.

To have a physically realistic simulation, it is necessary to provide the model with realistic bedload transport rates to introduce reliable boundary conditions and physical modelling within the study area extent.

**L65: I think there is some relevant literature about braided morphodynamics modeling that is missing. I suggest considering Williams et al. (2013), Williams et al. (2016a), Williams et al. (2016b), Rifai et al. (2014), Gonzales de Linares et al. (2021). Comparison with these works would also be relevant in the discussion**

**section.**
We will add the proposed literature, which completes usefully the previous ones.
For braided morphodynamics modelling, the model performance can be provided by a specific indicator a the scale of the area of interest (Williams et al. (2013), Williams et al. (2016a), Williams et al. (2016b), Rifai et al. (2014), Gonzales de Linares et al. (2021).

**Study area description: I think it would be of interest for the reader to add a more detailed description of the site, in particular, including a river profile showing slopes, the tributary, solid material input, etc. If available, I suggest to use a hillshade map of the catchment in Fig.1 (instead of the grey areas) to better show the relief and the mountainous environment. Maybe also add pictures illustrating the upstream main tributary morphology?**
A better description of the study site including length and slopes will be added in the text. A plot of the longitudinal profiles extracted from the LIDAR DEM of 2016 and 2019 will be plotted in the description part (Fig. 1).

**Fig2: I suggest writing directly on the picture the dates and mention "before the 2013 flood", "after the 2013 flood".**
This will be taken into account in the last version of the article.

**L104: "compared to 90 m3/s annually in the same period" -¿ do you mean: compared to the monthly averaged discharge of 90m3/s?**
Yes exactly. The sentence will be modified as proposed.
compared to the monthly averaged discharge of 90 $m^3/s$.

**2.4 Restoration implications: I have a concern regarding the model aim. I am not convinced that once calibrated on a rare event such a the 2013 event, the model will automatically be applicable for long term modeling, in particular for low magnitude events that would likely contribute to transport the stored sediments from the LDG to the downstream reaches in case the weirs are removed.**
The calibration was made on the 2018 event, which is a 10 year return period episode and not on the 2013 event, which is more extreme. We also chose this event because we consider that it can be more representative of the sediment transport phenomena that we want to consider for the restoration measures.

**L128: I would add something like "Thus, to help decision makers, a hydro-morphological 2D model..."**
This sentence will be added in lines 128-129.

**3.2.1 Friction laws: specify that the friction law is not coupled with the bed surface grain size (which is constant because fractional transport is not considered in the morphodynamical modeling).**
The friction law is indeed not coupled with the transport model. The value of $D_{84}$ is not calibrated but taken from measurement in the field. We will add the following sentence:
The value of $D_{84}$ is directly obtained thanks to the grain-size measurements done at the bed surface in the LDG area (see part 4.1.1).

**L170: I wouldn't say "The friction coefficient for the Ferguson (2007) law is the D84", I would say something like "the Ferguson law uses the D84 as a proxy of the bed roughness" or remove this sentence**
We decided to remove the sentence because the information appears clearly in equation (3).

**L185: I found the sentence not clear, K/K' is used for shear stress partitioning if I'm right. Do you use this shear stress partitioning in your 2D modeling? According to Gonzales de Linares et al. (2020, 2021), such correction might not be relevant in massive bedload transport. The recking formula: I have a major concern here. You use a 1D "morphological" formula that takes implicitly into account the cross-section averaging (variability of shear stress, grain size distribution, etc.) to deal with non-linearity effect of bedload transport. In your 2D model, the shear stress is spatially discretized so that the Shield number used to calculate the bedload flux is a "local" Shield number and the bedload transport formula does not need to take into account cross section averaging. I suggest considering the following papers: Recking (2013) and Recking et al. (2016).**

The MPM law was used with the correction because the conclusion of Gonzales et al. (2020, 2021) was not known during the simulation studies. But this will be mentioned in the discussion. The Recking formula that has been implemented is a version compatible with 2D calculation. It will be added in the text.

We used the version of this formula compatible with 2D calculation and local data (Recking et al., 2016a). It can be written as follows (Eq. 1):

$$q_b^* = \frac{q_b}{\rho_s \sqrt{g(s-1)D_{84}^3}} = 14 \frac{\tau^{*\,2.5}}{1+\left(\frac{\tau_m^*}{\tau^*}\right)^{10}} \tag{1}$$

$q_b^*$ $[-]$ is a dimensionless bedload discharge, $q_b$ $[kgs^{-1}m^{-1}]$ is the unit bedload discharge per unit width, $s = \rho_s/\rho$ is the specific gravity, and $g$ the gravity acceleration. $\tau^*$ $[-]$ is the Shield number, calculated from the diameter $D$: $\tau* = \frac{\tau}{g(\rho_s-\rho)D}$ with $\tau$ $[N/m^2]$ the shear stress. Here the calculations were made using $D_{84}$ as the grain diameter. The parameter $\tau_m^*$ is a mobility term that defines the transition between partial transport ($\tau^* < \tau_m^*$) and full mobility ($\tau^* > \tau_m^*$) (Recking et al., 2016a). The Recking formula was calibrated on field data ($\tau^* < \tau_m^*$) and laboratory data ($\tau^* > \tau_m^*$). It is the value of $\tau_m^*$ that gives its shape to the model. Therefore the value of $\tau_m^*$ strongly impacts the result, and its determination is difficult, especially for mountain streams. Ideally it should be based on measurements. Failing that, the available data suggest that an estimate is possible using Eq. 2 (Recking et al., 2016a).

$$\tau_m^* = 0.26S^{0.3} \tag{2}$$

**3.2 Sediment transport and bed evolution module: Could you specify if an avalanche mode (Slope sliding) is used and what are the parameter considered (angle of repose)? Same question regarding the deviation of sediment transport on transverse slopes? Both could have significant effect on the morphological modelling, it is thus of interest for the reader to know which parameters were used.**

The slope sliding effect was of course tested to analyse its influence on sediment transport. The two available formulae were considered. However, they both exaggerated sediment depositions between the two weirs. Hence, this parameter was not considered in the presented results. Besides, as the "angle of repose" option is only activated if one of the two slope sliding equations is used, it was not considered in the presented results but was analysed during the sensitivity analysis that we performed before the selection of the most relevant parameters.

**L223-224: could you provide the point density of the Lidar and the raster resolution of the used DEMs? Could you specify here if bathymetric data for the low flow channels were available?**

We will specify the resolution of the DEM in the last version of the article. The planimetric resolution is 1 meter. No reliable bathymetric data was available for 2019, thus we prefered to

apply the same methodology for 2016 and 2019. The two DEMs were surveyed during very low flow periods, which reduces the uncertainties concerning the bathymetry of the water area.

**L235: It is not clear for me how you estimate the bedload fraction? Do you have GSD measurement of the dredging? This should be clarified. The range 8-16% seams possible but much wider range could be considered, see for instance Turowski et al. (2010). I also wonder if you can consider the same partitioning between the pre and post flood situation: Fines will likely deposit more easily in a lake (even if they will deposit in the braiding situation). Note that the sediment transport partitioning might also evolved with the event return period and the flow/sediment supply conditions so that dredging might not be fully representative of a specific event. The following papers could be of interest to consider a gravel matrix fine fraction commonly observed in gravel bedded stream: Mueller and Pitlick (2013), Navratil et al. (2010), Misset et al, (2020)**

The dredging data were used to estimate the bedload fraction. However, no GSD measurements of the samples were available. This will be clarified in the last version of the article. The model presented in the article considers only one sediment diameter: $D_{84}$. Therefore, it does not allow to check the interesting remark regarding the evolution during the flood. To overcome this drawback, simulations on the updated sediment transport module GAIA, developed to handle the grain-size distribution issue better, are currently in progress.

**4.1.2 Input hydrograph: do you have points with liquid discharge measurement to calibrate/validate the model? This should be specified.**

We will add the following information to explain the use of the hydrological model MARINE. This model has been calibrated based on the available observed discharges at three stations: the Gave de Cauterets, the Gave de Gavarnie and the Gave de Pau after the confluence with the Gave d'Azun. 6 events extracted from these observed time-series allowed calibrating the model with a good confidence.

**Fig5. It is not clear for me if the large model uses a fixed bed? I suggest to add a title above each mesh (large, finer mesh).**

It considers non erodible areas but the bed is not fixed. We will add the following information: For the simulation with both meshes, the sediment transport model was used.

**L277: the equilibrium load upstream condition: Is this relevant? Were the upstream profile and cross section stable? Did you compare these upstream fluxes with an averaged 1D formula? This hypothesis is probably critical for the modeling of the event.**

We agree with these comments that are very relevant. The simulation in the first meters upstream is not considered to be relevant as the simulated fluxes do not match the ones estimated with the 1D reach averaged formula. However, when we use the larger model we see that the bed evolution is not important at the upstream boundary of the smaller mesh (see also response to general comments).

**L285: I suggest specifying that under the hydrodynamics calibration conditions, morphological changes and bedload transport were limited. If bathymetric data were not available, this can probably lead to a non-negligeable uncertainty on the water surface elevation, which can be considered acceptable compared to other uncertainties, this need to be specified. Also, did you vary the D84 considered in the Ferguson friction law? Or did you use the measured value?**

These relevant remarks will be integrated in the last version of the manuscript. The uncertainty

due to the lack of bathymetry can be estimated considering the water depth during the LIDAR survey which was approximately the same for both DEMs. This will be specified in the last version of the article.

We didn't vary the $D_{84}$; we only used the measured value.

**L305-306: maybe use the term DoD (DEM of differences)?**
The term DoD will be used.

**4.4 Performance evaluation: Braiding and morphological changes in braided rivers are somewhat stochastic processes. Is it thus relevant to use single long profile/cross sections and the Brier Skill Score objective function? I mean, your model can be considered relevant in reproducing the morphodynamics of the studied reach as it reproduces well some general morphological evolution (for instance the braiding index, the number of active channels, the statistical distribution of erosion and deposition, the average long profile, etc.) but the objective function can lead to a bad score (you do not have exactly the same DEM evolution).**
We totally agree with this remark and we tried to criticize the use of the BSS and long profile evolution in the discussion section. Besides, since morphological changes are indeed a stochastic phenomenon in braided rivers, we considered that a volumetric analysis in which we compare the deposition volumes might me more relevant and sufficient for the purpose of our study. We will add more precision in the discussion section and introduce other more relevant criteria to consider for the analysis of morphological changes in braided rivers (such as the ones that you suggest by Williams et al. (2016b)).

**4.4.3 Comparison of the deposited volumes Comparison of the global volumetric budget of the reach is of great interest. I have however the filling that DoD obtained for the 2018 flood includes much more information. You could for instance compare that statistical distribution of erosion/deposition in the reach, is is done in a similar study by Williams et al. (2016b).**
Thank you very much for this suggestion. The statistical distribution of erosion/deposition will be analysed upstream the LDG (Fig. 2) and within the LDG (Fig. 3). We observe that the Strickler friction equation has a completely different dynamic and it tends to exagerate sediment depositions compared to the Ferguson formula. These two figures will be added and commented in the last version of the article.

**Fig7: I suggest using red for erosion and blue for deposition (commonly used colors for such map), to include a scale bar and to remove the pixel within the uncertainty range of the DoD (by using transparent color so that only significant morphological differences can be seen).**
We will modify this figure according to your suggestions.

**L349: Friction laws link velocity to depth and roughness**
The sentence will be modified.

**5.1 General visual comparison of eroded and deposited areas: according to a previous comment, you are comparing visually the erosion and deposition distribution depending on the friction law and transport formula used: comparing these distributions could strengthen this analysis. Another aspect that could be discussed regarding the effect of the friction law is the use of spatially and temporally constant roughness parameters (n or D84). It is likely that these parameters are not constant in both space and time the braiding reach, which can explain some of the differences between modeled and measured evolutions.**
Indeed, in this section we intended to compare the visual erosion and distribution depending on

[Figure]

Figure 2: Statistical distribution of erosion/deposition observed with the DoD compared with the simulated evolution upstream the LDG

both the friction law and the sediment transport equation.

We will add a comment regarding the effect of the roughness parameters to better discuss the impact of uniform and constant $K$ or $D_{84}$.

In line 380 : Besides, it is likely that the roughness parameters used by the two considered friction laws ($K$ or $D_{84}$) are not constant in both space and time. This can also explain some of the differences between modeled and measured evolutions.

**L372: If I understand, the model seems to be not fully relevant in the downstream part of the reach, as almost no evolution is modeled. I agree that such model, already complex, doesn't take into account all the complexity of such braided system. I however wonder if this disagreement is only due to the fact that suspended load is not modeled. Do you have observations showing that the downstream part is mainly driven by suspended load processes? Based on aerial photograph, the downstream part of the reach seems to be at least partially composed of gravel bars? I suggest moving this discussion of the results in the discussion section.**

This part of the reach is mainly affected by suspended load even if we can see some gravel in aerial photos. In fact we performed geotechnical surveys (- 4m) in this area to analyse the composition of the deposited materials and gravel is only observed within the first 20 cm. Once this surface layer is passed we only observed very fine and cohesive sediments such as silt. This is why we supposed that this part of the reach is mostly affected by suspended load.

We modified these two sentences to explain our assumptions:

This can be due to the fact that many factors are not considered in the model such as the consideration of the whole grain-size distribution for bedload.

and the following one

The morphological evolution within this section is thus, for the most part, probably due to suspended load, not considered in our model.

[Figure]

Figure 3: Statistical distribution of erosion/deposition observed with the DoD compared with the simulated evolution within the LDG

**Fig.8 and 9: I suggest adding on the map "observed changes" and the names of the transport formula and friction law.**
This will be added in the figures.

**5.2 Longitudinal profiles and cross-section comparison: I wonder if comparing long profile in braided morphology is relevant? Maybe, using an "averaged" long profile to capture the general trend and being less sensitive the the active channel position would be a better option? I also suggest to move from this section all sentences discussing this specific point and the relevance of the BSS index.**
To be consistent with the other part we did not use transversal averaging to analyse local evolution. Obviously, this method does not give an exhaustive view of bed modification. But in this first study, we considered that volume analysis can provide sufficient results.
However the addition of the statistical distribution of erosion and disposition can also help and will be added (see general comment n°5 about the performance criteria and Fig. 2 and 3).

**L415: Can we consider the deposited bedload fraction is similar between the 2018 braiding situation and the pre 2013 flood situation with a lake? I have the feeling that the deposited suspended load fraction would be higher in a lake configuration. Were the hydrodynamic conditions low enough (backwater effect, low velocities, low shear stress, etc.) during the 2018 flood so that more than 80% of the deposited volume corresponds to suspended particles? If it is not the case, this fraction seems to be really high. I suggest comparing it with previous works on fine particle stocks/content in gravel bedded streams (see Mueller and Pitlick (2013), Navratil et al. (2010), Misset et al, (2020)).**
This point has been discussed in a previous comment. Thank you for the additional references that will be added in the last version of the article.

**References**

Gonzales de Linares, M., Mano, V., Piton, G., and Recking, A.: Modelling of massive bedload deposition in a debris basin: cross comparison between numerical and small scale modelling, in: RiverFlow 2020, edited by et al., U., Proceedings of the 10th Conference on Fluvial Hydraulics, Delft, Netherlands, URL `https://hal.archives-ouvertes.fr/hal-02935173`, 2020.

Gonzales de Linares, M., Ronzani, F., Recking, A., Mano, V., and Piton, G.: Coupling Surface Grain-Size and Friction for Realistic 2D Modelling of Channel Dynamics on Massive Bedload Deposition, in: Conference: SimHydro 2021: Models for complex and global water issues - Practices and expectations, vol. 30 of *Actes de la conférence SimHydro 2021*, pp. 1–10, Sophia-Antipolis, France, URL `https://hal.archives-ouvertes.fr/hal-03363327`, 2021.

Misset, C., Recking, A., Legout, C., Viana-Bandeira, B., and Poirel, A.: Assessment of fine sediment river bed stocks in seven Alpine catchments, CATENA, 196, 1–14, https://doi.org/10.1016/j.catena.2020.104916, 2021.

Mueller, E. R. and Pitlick, J.: Sediment supply and channel morphology in mountain river systems: 1. Relative importance of lithology, topography, and climate, Journal of Geophysical Research: Earth Surface, 118, 2325–2342, https://doi.org/https://doi.org/10.1002/2013JF002843, 2013.

Navratil, O., Legout, C., Gateuille, D., Esteves, M., and Liébault, F.: Assessment of intermediate fine sediment storage in a braided river reach (southern French Prealps), Hydrological Processes, 24, 1318–1332, https://doi.org/10.1002/hyp.7594, 2010.

Recking, A.: An analysis of nonlinearity effects on bed load transport prediction, Journal of Geophysical Research: Earth Surface, 118, 1264–1281, https://doi.org/https://doi.org/10.1002/jgrf.20090, 2013.

Recking, A., Piton, G., Vazquez-Tarrio, D., and Parker, G.: Quantifying the Morphological Print of Bedload Transport, Earth Surface Processes and Landforms, 41, 809–822, https://doi.org/10.1002/esp.3869, 2016a.

Recking, A., Piton, G., Vazquez-Tarrio, D., and Parker, G.: Quantifying the Morphological Print of Bedload Transport, Earth Surface Processes and Landforms, 41, 809–822, https://doi.org/10.1002/esp.3869, 2016b.

Reisenbüchler, M., Bui, M. D., Skublics, D., and Rutschmann, P.: Enhancement of a numerical model system for reliably predicting morphological development in the Saalach River, International Journal of River Basin Management, 18, 335–347, https://doi.org/10.1080/15715124.2019.1628034, publisher: Taylor & Francis, 2020.

Rifai, I., Le Bouteiller, C., and Recking, A.: Numerical study of braiding channels formation, in: Telemac Mascaret User Club Conference, pp. 159–167, Grenoble, France, URL `https://hal.inrae.fr/hal-02606139`, 2014.

TUROWSKI, J. M., RICKENMANN, D., and DADSON, S. J.: The partitioning of the total sediment load of a river into suspended load and bedload: a review of empirical data, Sedimentology, 57, 1126–1146, https://doi.org/https://doi.org/10.1111/j.1365-3091.2009.01140.x, 2010.

Williams, R. D., Brasington, J., Hicks, M., Measures, R., Rennie, C. D., and Vericat, D.: Hydraulic validation of two-dimensional simulations of braided river flow with spatially continuous aDcp data, Water Resources Research, 49, 5183–5205, https://doi.org/https://doi.org/10.1002/wrcr.20391, 2013.

Williams, R. D., Brasington, J., and Hicks, D. M.: Numerical Modelling of Braided River Morphodynamics: Review and Future Challenges, Geography Compass, 10, 102–127, https://doi.org/https://doi.org/10.1111/gec3.12260, 2016a.

Williams, R. D., Measures, R., Hicks, D. M., and Brasington, J.: Assessment of a numerical model to reproduce event-scale erosion and deposition distributions in a braided river, Water Resources Research, 52, 6621–6642, https://doi.org/https://doi.org/10.1002/2015WR018491, 2016b.

---

## Author Comment (AC3)

**Answer to RC3: Comment on esurf-2021-91 from Saraswati Thapa**

July 3, 2022

Dear reviewer,

Thank you very much for this detailed and constructive review which will be very useful to clarify and improve our article. Please find below the detailed answers. The reviewer's comments are shown in bold and some modifications of the manuscript are emphasized in blue.

**Review of "Numerical modelling of the evolution of a river reach with a complex morphology to help define future sustainable restoration decisions", submitted to Earth Surface Dynamics, January 2022.**

**Dear editor and dear authors,**

**This review has been done jointly by Saraswati Thapa and Mikael Attal. We have read this manuscript with great interest: the research topic is very valuable in its content and we like that the paper highlights the importance of the numerical modelling approach for the evolution of a river reach in response to extreme flood events. We need more studies such as this one, that do combine high resolution pre and post surveys with numerical modelling, to test and calibrate models, and to assess their ability to replicate a range of features of importance to scientists and policy-makers, e.g., volumes eroded and deposited, changes in elevation and morphology, predicted response to anthropogenic changes (land use or risk management).**

**We enjoyed reading this manuscript that presents a very nice set of experiments using the TELEMAC-MASCARET model to reproduce the dramatic changes that occurred in a reach of the Gave the Pau. The results are enlightening, providing answers to a series of scientific questions and directions for future work. However, there are issues that need to be addressed before publication.**

**One of the main issues is the weak motivation for using this model in this particular example. We feel this could be better motivated, in particular when it appears, as we go through the results, that this model is not very good at reproducing braiding or suspended sediment transport (and this is a braided reach with ~90% sediment transport in suspension!) There are many landscape evolution models considering many sediment transport equations and multiple grain sizes available. The model in this study used two bed load transport equations and neglected suspended load. The study area has very heterogeneous grain size, however, the model used single grain size D50 for the MPM formula and the D84 for the Recking formula rather than multiple grain size distribution (see for example Ramirez, J. A. et al. (2020) 'Modeling the geomorphic response to early river engineering works using CAESAR-Lisflood', Anthropocene, 32. doi:10.1016/j.ancene.2020.100266).**

**We made recommendations in the annotated manuscript, and one of the suggestions is that you could highlight the strengths and weaknesses of the model from the onset, highlight that the strengths make this model a good potential candidate to model the changes in the study area (it is one of a few models available that are**

able to model morphodynamics (erosion and deposition) during large flood events), and that here you use this well constrained example to assess the model's ability to reproduce volumes and cross-sections, and assess its suitability as a tool to inform policy makers" (or something along these lines). Having a clear motivation for the use of the model and clear aims will strengthen the argument, as the reader will know what to expect as they progress through the paper. You can also build on these aims to justify the strategy for modelling, that is, which parameters you are planning to test (or not) and why. The information is there in the text, but we feel it would help if that were presented clearly at the onset. It is also important to build the argument on literature, and we have made suggestions throughout the text.

In general, the paper could do with more details in many sections: description of the model and parameters, description of how data were collected, description of results. There are also places where the outcomes of the model could be evaluated in a more quantitative way. This is particularly crucial for the last section where the impact of restoration scenarios is assessed through a couple of cross-sections, when the previous section demonstrated that the model was not very good at modelling cross-sections and better at modelling volumes.

Thank you very much for these helpful suggestions that we have all taken into account and will for sure improve the quality of our article.

A better description of the study site, including length and slopes, will be added to the text. We will also add a description of the data collection methods. A plot of the longitudinal profiles extracted from the LIDAR DEM of 2016 and 2019 will be added in the description part so that the reader better understands the characteristics of the study area. The figure that we intend to add to the last version of the article is presented below (Fig. 1).

[Figure]

Figure 1: Longitudinal profile evolution following the flood of 2018.

Regarding the assessment of the restoration scenarios, we intended to show the riverbed

elevation, which can increase flood risks, and not the exact braiding and morphological changes that are stochastic processes. We will, however add a volumetric analysis to be consistent with our discussion (section 5.4).

**Finally, the writing can be improved. We have made suggestions in the attached annotated manuscript.**
Thank you very much for these helpful suggestions that we have all taken into account. You will find below the details of the answers, except for the language corrections which were made directly in the manuscript without mention in this document.

**This is an original study. Very few studies have attempted to apply numerical models to natural / real examples to model morphological changes on these space and time scales. We believe there is potential for a strong publication. We hope you find these suggestions useful and wish you all the best with your revisions.**

**Comments in the annotated manuscript:**

**L36: There have been a few publications with the CAESAR-Lisflood model as well in the past two years, which may be relevant? See for example: Ramirez, J. A. et al. (2020) 'Modeling the geomorphic response to early river engineering works using CAESAR-Lisflood', Anthropocene, 32. doi:10.1016/j.ancene.2020.100266.**
Thank you for this relevant reference that has been studied and added to the last version of the article.

**L63-64: How is this model considered to be suitable for this study as it has not been explored much? That is a very important point, in particular since one of the outcomes of the study is that the model may not be suitable due to not modelling suspended sediment. You may want to rephrase by describing the model, its strengths and weaknesses, and explaining that this study is a test of the model in a setting where it would be expected to perform well (i.e., it is developed to model sediment transport at the event timescale and is suited to areas with large sediment fluxes).**
Clarifications will be added to the text.
Indeed, previous studies have shown that TELEMAC/Sisyphe was able to reproduce processes of erosion/deposition accurately in similar configurations (Reisenbüchler et al., 2020, 2019; Cordier et al., 2019). Sisyphe enables the use of different transport formulas (Meyer-Peter and Müller, 1948; van Rijn, 1984) and also take into account various factors influencing sediment transport, such as the effect of the bed slope (Koch and Flokstra, 1981; Soulsby, 1997) on the magnitude of the bedload transport (Riesterer et al., 2016). It also offers the possibility of programming other formulas, both for the parameterisation of friction and for solid transport, a possibility which has been used here to introduce formulations more adapted to the context of mountain rivers.

**L182-188: What is $\Phi$ here? Is this K'/K or K/K' please double check. Also, it is not clear how D* should be grain size for grain class i. Can we correlate it with D50 or D90? Could you please add the referencing for this formula as well?**
The definitions of $\Phi$ and $\tau*$ have been added in the manuscript.
The formula for the estimation of the grain roughness coefficient $K'$ was also established by Meyer-Peter and Müller (1948) and involves the $D_{90}$, it has been clarified in the text.
$\Phi$ is the dimensionless solid transport, calculated as $\Phi = \frac{q_{sv}}{\sqrt{g(\rho_s/\rho-1)D^3}}$ with $q_{sv}$ $[m^3/s/m]$ the unit solid volume transport: $q_{sv} = Q_{sv}/W$ with $Q_{sv}$ $[m^3/s]$ the solid volume flow rate, $W$ $[m]$ the river width, $\rho_s$ $[kg/m^3]$ the density of the sediments, $\rho$ $[kg/m^3]$ the density of water, $g$ the

gravity acceleration and $D$ $[m]$ the grain diameter. $K/K'$ is the ratio between the flow Strickler coefficient $K$ and the grain roughness coefficient $K'$. This term makes it possible to correct the total constraint in order to take into account only the grain shear stress. $K$ is given by $K = \frac{U}{S^{1/2}R^{2/3}}$ and according to Meyer-Peter and Müller (1948) the grain roughness coefficient can be estimated as a function of the grain size distribution $K' = \frac{1}{n} = \frac{26}{D_{90}^{1/6}}$, with $D_{90}$ the diameter at about 90% by weight of the grains [m]. $\tau*$ $[-]$ is the Shield number, calculated as $\tau* = \frac{\tau}{g(\rho_s - \rho)D}$ with $\tau$ $[N/m^2]$ the shear stress.

**L190: What is that threshold value?**
The threshold value characterize the incipient motion of sediment. Clarifications will be added to the text.
The Meyer-Peter-Müller equation is an excess shear relationship and its original formulation considers a critical Shields parameter equal to 0.047 as a threshold for characterizing the incipient motion of bed grains.

**Eq. 6: What is $q_b^*$?**
Clarifications will be added to the text.
$q_b^*$ $[-]$ is a dimensionless bedload discharge.

**Eq. 6: What is $\tau_{84}^*$?**
Clarifications will be added to the text.
$\tau_{84}^*$ $[-]$ is the Shield number, calculated from the diameter $D_{84}$: $\tau* = \frac{\tau}{g(\rho_s - \rho)D_{84}}$ with $\tau$ $[N/m^2]$ the shear stress.

**L209: 'qsform.f': Make available?**
Of course, fortran routines can be made available on request.

**L231: Give more details. No spatial variability in grain size?**
Only localized data have been collected over several sediment bars.

**Figure 4: There are four sampling locations mentioned so what are those C4 and G3 locations for? Also, could you georeference the figure?**
C4 and G3 are the grain size distributions on the Gave de Cauterets and the Gave de Gavarnie, upstream tributaries of the Gave de Pau. The figure will be georeferenced.

**L235: Show data or cite source.**
The data is presented in the figure below (Fig. 2). It has been communicated by the former hydropower company that was in charge of the exploitation of the two weirs before the flood of 2013.

**L236-238: It means a very high suspended load. So you need strong justification why have you chosen this model, which does not consider suspended load?**
**L375-376: This location has high suspended load so you need to better justify the use of this model without suspended load.**
**L485-486: See earlier comments - need to justify the use of the model in this context, since it appears unable to model sediment in suspension.**
In the Lac des Gaves, the deposit consists of an upper layer of coarse material over the first few centimeters, with finer sediment stored below and these fine sediments were mainly deposited downstream of the Lac des Gaves. The coarse sediments constituting the upper layer, it is the choice of a simulation of the bedload transport which was made.
Besides, bedload was the main concern of the river managers in the studied area since it is the fraction that controles the stability of the channel and is at the origin of the observed impacts such as erosions and depositions.
This will be better explained in the article.

[Figure]

$$V_{mean} = 2\,760\,m^3/year$$

Figure 2: Longitudinal profile evolution following the flood of 2018.

**L237: observation? Data?**
This information comes from the available dredging data (5years).
This will be specified in the article.

**L241-244: Which data are used to constrain the scenarios: discharge gauges? Rainfall gauges?**
The hydrological model needs several input data, clarifications have been added in the text.
The data used for implementing MARINE model include rainfall (source: Météo France), topography (source: IGN), soil properties (source: INRA), land use (source: CORINE Land Cover) and event discharge (source: HydroEau France (DREAL)  EDF).

**L248-250: Could you also add the name of these tributaries on the figure as well?**
**Figure 5: I suggest adding the name of tributaries, georeferencing, and Scale bar.**
This will be done.

**The tenses are changing a lot from present to past to present (transition to next section). Please ensure consistency.**
This will be done.

**L260-261: A fine mesh should take more time to compute. How can it be less time consuming? Please double check.**
Clarifications will be added to the text.
The finer mesh covers a much smaller area, so it is used to perform a less time consuming fine analysis of the sediment transport behaviour around the area of interest

**L285: Did you measure the discharge yourself? If not, please mention the source.**
No the discharge was measured by a public service in France named DREAL. This will be specified in the text.

**L287-288: What is the significance of choosing this range of values?**
This is a classical range of values for roughness in natural river.

**Figure 6: Locate the weirs?**
This will be done.

**L295: Not entirely clear: what do you mean?**
We meant that there are no other topographic campaigns between 2016 and 2019 that would allow the observation of the effects of the 2018 flood only. Clarifications will be added to the text.
Unfortunately, there have been no topographic campaigns between 2016 and 2019 that would account for the effects of the 2018 flood only.

**L296-297: Show what you've got and your results before saying "more work is needed".**
The sentence has been removed.

**L325-328: Are they? Many models have been developed that do try to do that. They are not necessarily doing a great job, but there are numerical models of braided system, they exist. Are you referring to the fact that we cannot exactly model where a channel or a bar will be? That is certainly true. Scientists have been developing metrics to assess the goodness of a fit in such context without just using the ability to geographically match the location of features, see for example the work by HAjek and colleagues, and for example this paper that looks at clustering of sand bodies in stratigraphy (this is not for braided system but just to illustrate the point): https://pubs.geoscienceworld.org/gsa/geology/article/38/6/535/130307**
**See also https://www.sciencedirect.com/science/article/pii/S0037073811002260#section-cited-by**
Yes, we were refering to the ability to geographically match the location of channels or bars. Thank you for the references, clarifications will be added to the text. On the suggestion of another reviewer, we have added another metric: the statistical distribution of erosion and disposition (Williams et al., 2016).
The statistical distribution of erosion/deposition has been analysed upstream the LDG (Fig. 3) and within the LDG (Fig. 4). We observe that the Strickler friction equation has a completely different dynamic and it tends to exaggerate sediment depositions compared to the Ferguson formula. These two figures will be added and commented in the last version of the article. it can be seen that some models reproduce the spatial distribution of deposited or eroded areas better than others, this will be further discussed in the article to improve the analysis.

**Figure 7: Is that modelled? Measured? Say this is a DoD?**
Yes, this is a DoD, Field erosion and deposition areas were estimated through topo-bathymetric differencing between two LiDAR DEMs surveyed in 2016 and 2019. Clarifications will be added to the legend.
Eroded (light blue) and deposited (dark green) areas in the LDG reach estimated through topo-bathymetric differencing between two LiDAR DEMs surveyed in 2016 and 2019

**L346-348: I think figure 'a' is more matching with d and e rather than b and c? Is there a quantitative way of comparing? Can you compare the volume of erosion**

[Figure]

Figure 3: Statistical distribution of erosion/deposition observed with the DoD compared with the simulated evolution upstream the LDG

**and deposition for all of these cases? I imagine you have used the same color level for the all figures? Regarding this statement, the magnitude of change seems to be greater in b and c than in d and e, whereas the extent of the deposition area is greater in d and e. Describe these results, similarities and differences in greater detail?**

**Figure 8: From these figures, I think figure a more matching with c and d rather than a nd b? I hope you have used the same color level for the all figures.**

The same color levels have been used for all the figures, it's the one specified in the colorbar on the left.

Yes you're right figure 8(a) is more matching with d and e rather than b and c. It's probably due to the fact that the Strickler formula seems to overestimate erosion and deposition processes. Since the simulations only take into account bedload and the difference in DEMs obviously represents total load, it is therefore logical that the results of the formula overestimating the bedload seem closer to the observations of the total load. Of course this is only a qualitative way of comparison. The volume of erosion and deposition by bedload only for all these cases are compared later in the text (§5.3, table 2).

The similarities and differences will be more detailed in the last version of the article.

**L351: Can you define the transition from low to high submergence? Would you consider the floods you model in the LDG as low or high submergence?**

Low submergence corresponds to a water height of the same order of magnitude as the roughness. At the beginning of the flood, the submergence is low then it becomes high during the peak flood. Clarifications will be added to the text.

whereas the Ferguson friction law is known to have the best performance from low to high submergence (Rickenmann and Recking, 2011) which is probably more suited to our case study, for which the water height is of the same order of magnitude as the roughness at the beginning of

[Figure]

Figure 4: Statistical distribution of erosion/deposition observed with the DoD compared with the simulated evolution within the LDG

the flood. Then the submergence becomes high during the peak flood.

**L357: Why should it overestimate?**
MPM is a threshold formula. However, the simulation of the shear stress is subject to uncertainty, so around the threshold, if the simulated shear stress is underestimated there will be no sediment transport and if it is overestimated, there will be simulated sediment transport when there was none in reality. Therefore, the computational uncertainty around the shear stress threshold results in greater uncertainty in the sediment transport.

**L373-374: Have you explored the multiple grain-size distribution option and is there any limitation to go with multiple grain size distribution?**
The multiple grain-size distribution option option was not available in the solid transport module Sisyphe with the formulas used but work is underway to test its impact with the new solid transport module Gaia of the TELEMAC-MASCARET modelling system.

**L378: How do you produce a longitudinal profile in a braided system?**
The 1D plot is just an extraction of the longitudinal profile from the lowest bathymetric points of the 2D results. Clarification will be added in the text.
The 1D longitudinal profiles of the present paper are drawn from an extraction of the lowest bathymetric points of the 2D model.

**Figure 8: Add labels next to b-e to facilitate the reading? e.g., MPM-Ferg, Reck-Ferg, MPM-Strick...**

This will be done.

**Figure 9: The results look quite different compared to Fig. 8b and c - the patterns don't exactly match. Why is this the case?**

The results are the same on the common area located donwstream of the upstream weir but it's not the same color levels that have been used for figures 8 and 9.

**L387-389: Do you think it would be possible to improve the performance with some additional simulations, for example by widening the input parameter like roughness? L389: Or the ability of the model to replicate the processes operating in this catchment.**

We made several additional simulations that are not presented in the manuscript for the sake of simplicity. However the poor BSS values are mainly due to the fact that the model is not able to geographically match the location of channels or bars which is a known weakness of hydrosedimentary models. Clarifications will be added to the text.

This questions the relevance of the BSS criteria for complex morphologies such as the braided LDG reach as the hydrosedimentary model cannot simulate exactly where a channel or a bar will be.

**L390-391: I don't understand: this looks pretty good to me! I can see across-channel relief that implies bars and multiple channels. What is so bad about this simulation?**

The sentence will be reformulated as follows.

As mentionned above, Figure 13 shows that the model experiences difficulties to geographically match the location of channels and bars forming the braided morphology of the LDG reach.

**L393: How much of this erosion is associated with the migration of bars, which would look like erosion next to deposition without a net change in sediment volume? Show the original 2016 profile for comparison on the figure?**

This is a very interesting question. The sediment volume will be estimated in this area to verify if the observed erosion is associated with the migration of bars or if the mobilized materials were purged downstream.

**L400: But this is not the only thing that matters, is it?**

Of course mobilised sediment volumes are not the only important subject. However a knowledge of the mobilised volumes would already allow to progress in the elaboration of the knowledge of the site and the proposal of restoration project. It would already allow to estimate, for example, what volume of sediments could go downstream in a scenario of removal of the downstream weir.

**Figure 10: It was said earlier in the text that the upstream condition was one where the elevation of the inlet does not change (line 277)**

Yes, this is true, the upstream boundary condition imposes a constant elevation at the inlet. As it can be seen on the left part of the figure, the first measurement point is not located at the inlet, that's why the elevation changes. Clarification will be added in the legend of the figures.

Attention, the first point of the longitudinal profile is not located at the inlet.

**L408-410: This is something that should have stated at the very beginning - this is something you can use to justify the study and use of the model. By making this statement from the onset, you can justify the adoption of a given modelling strategy.**

The sentence has been added at the beginning, in §2.4 Restoration implications.

One of the processes on which the modeling efforts will focus is the deposition phenomenon within the LDG as it represents the potential volumes that might be mobilised if the weir lowering/removal restoration measure is considered.

**L411-412: Can you give more details? Were the volumes estimated through DoD?**

**L413-414: Is the sediment deposition volume estimated from DEM of difference or any other approach, not mentioned here?**

The total volume was estimated through DoD. The bedload fraction was estimated to represent 8 to 16% of the total observed deposited volume. This percentage value comes from the dredging data.

**Table 2: Here, In the table, the simulated bedload volume score value in the lower limit is higher than the upper limit, could you explain it?**

The lower limit represents 8% of the total observed deposited volume. The simulated bedload volumes are much higher than this lower limit, they are closer to 16% which is the upper limit. That's why the score is better for this upper limit ($r$ closer to 1).

**Figure 13: Could you use different color for showing cross-section in plan as you are showing the same green color for simulated bed level so just to avoid the confusion.**

This will be done.

**Figure 13: How were the cross-sections chosen? You may show a couple more to illustrate the variability as you move away from the weirs?**

Many cross-sections were analysed during the post-processing of the results. However, to not overload the article, only the ones around the weirs (where most sediment transport processes occur) are presented.

**L436-437: So, why was this model chosen? It would be good to develop a better justification at the onset of the paper ("the model is not developed to reproduce braiding or deal with suspended sediment; however it is one of a few models available that are able to model morphodynamics (erosion and deposition) during large flood events. Here we use this well constrained example to assess its ability to reproduce volumes and cross-sections, and assess its suitability as a tool to inform policy makers".**

Thank you for this suggestion. The sentence has been added at the beginning, in §3 Model description.

The model is not developed to reproduce braiding or deal with suspended sediment; however it is one of a few models available that are able to model morphodynamics (erosion and deposition) during large flood events. Here we use this well constrained example to assess its ability to reproduce volumes and cross-sections, and assess its suitability as a tool to inform policy makers.

**L441: How can you be sure?**

The sentence will be rephrased.

Therefore another criterion adapted to braided rivers has also been considered: the statistical distribution of erosion and disposition (Williams et al., 2016). (see the answer at the beginning of this document).

**L450: I do not understand: in the last few sections, you explained that the model was not very good at getting the sections right and that it was better at estimating**

**volumes. Can you also estimate volumes and show DoD so that the reader can better visualise the outcomes?**

We understand the question which is relevant. The idea was just to have an idea of the morphodynamic processes (erosions and depositions) and have some orders of magnitude and not to have the exact location of the erosions/depositions. We will add a DoD to be consistent with the previous analysis.

**L454: why this choice?**

10-year return period-like flood events were chosen because they are both relatively large flood events with a rather low return period. Besides, this was also discussed with the river managers, who wanted to be prepared for such kinds of events that might occur more and more frequently. This will be explained in the article.

**L468-470: Can you check? I imagine you can calculate the shear stress during the event? You should be able to answer to this question with your data.**

Yes we are able to calculate the shear stress during the event. This will be checked and added to the last version of the article.

**L494-497: Rephrase / be more specific.**

The sentence will be rephrased.

One recommendation to decision makers is to not only consider the downstream weir but to consider both weirs in the restoration project.

**References**

Cordier, F., Tassi, P., Claude, N., Crosato, A., Rodrigues, S., and Pham Van Bang, D.: Numerical Study of Alternate Bars in Alluvial Channels With Nonuniform Sediment, Water Resources Research, 55, 2976–3003, https://doi.org/https://doi.org/10.1029/2017WR022420, 2019.

Koch, F. and Flokstra, C.: Bed level computations for curved alluvial channels, in: XIXth Congress of the International Association for Hydraulics Research; New Delhi, India, 1981.

Meyer-Peter, E. and Müller, R.: Formulas for Bed-Load transport, IAHSR 2nd meeting, Stockholm, appendix 2, publisher: IAHR, 1948.

Reisenbüchler, M., Bui, M. D., Skublics, D., and Rutschmann, P.: An integrated approach for investigating the correlation between floods and river morphology: A case study of the Saalach River, Germany, Science of The Total Environment, 647, 814–826, https://doi.org/10.1016/j.scitotenv.2018.08.018, 2019.

Reisenbüchler, M., Bui, M. D., Skublics, D., and Rutschmann, P.: Enhancement of a numerical model system for reliably predicting morphological development in the Saalach River, International Journal of River Basin Management, 18, 335–347, https://doi.org/10.1080/15715124.2019.1628034, publisher: Taylor & Francis, 2020.

Rickenmann, D. and Recking, A.: Evaluation of flow resistance in gravel-bed rivers through a large filed data set, Water resources research, 2011.

Riesterer, J., Wenka, T., Brudy-Zippelius, T., and Nestmann, F.: Bed load transport modeling of a secondary flow influenced curved channel with 2D and 3D numerical models, Journal of Applied Water Engineering and Research, 4, 54–66, https://doi.org/10.1080/23249676.2016.1163649, publisher: Taylor & Francis _eprint: https://doi.org/10.1080/23249676.2016.1163649, 2016.

Soulsby, R.: Dynamics of Marine Sands: A Manual for Practical Applications, Thomas Telford, London, 1997.

van Rijn, L. C.: Sediment Transport, Part II: Suspended Load Transport, Journal of Hydraulic Engineering, 110, 1613–1641, https://doi.org/10.1061/(ASCE)0733-9429(1984)110:11(1613), 1984.

Williams, R. D., Measures, R., Hicks, D. M., and Brasington, J.: Assessment of a numerical model to reproduce event-scale erosion and deposition distributions in a braided river, Water Resources Research, 52, 6621–6642, https://doi.org/10.1002/2015WR018491, _eprint: https://onlinelibrary.wiley.com/doi/pdf/10.1002/2015WR018491, 2016.

---

## Author Response (AR2)

**Athors response to the reviewers' comments**

October 30, 2022

Dear editor,

Thank you very much for giving us the opportunity to improve our article by integrating the comments that we received on our preprint.

All the received comments were very constructive, and most of them were incorporated to enhance the quality of our article. Please find below our detailed answers to the three reviewers. The reviewer's comments are shown in bold, and some manuscript modifications are emphasized in different colors for each reviewer.

In blue, the answers to the RC1.

In red, the answers to the RC2.

In magenta, the answers to the RC3.

We remain available to answer your questions if needed.

Kind regards,

The authors.

---

## Author Response (AR3)

**Authors response to the editor's last comments**

April 13, 2023

Dear editor,

Thank you very much for giving us the opportunity to improve our article by integrating the comments that we received on our preprint.
The last comments were incorporated in the last version of our article.
We remain available to answer your questions if needed.

Kind regards,
The authors.